# Exploring Health Promotion Behaviors, Occupational Burnout, and Sleep Disturbances in Traditional Industry Workers

**DOI:** 10.3390/healthcare13010051

**Published:** 2024-12-30

**Authors:** Ying-Fen Yu, Yi-Ya Chang, Shu-Hung Chang

**Affiliations:** 1Department of Nursing, Taoyuan Armed Forces General Hospital, Taoyuan City 325, Taiwan; m108013@mail.cgust.edu.tw; 2Department of Gerontology and Health Care Management, Chang Gung University of Science and Technology, Taoyuan City 333, Taiwan; 3Department of Nursing, Chang Gung University of Science and Technology, Taoyuan City 333, Taiwan; yychang@mail.cgust.edu.tw; 4Department of Health Management, Chang Gung Memorial Hospital, Taipei City 105, Taiwan; 5Department of Gastroenterology and Hepatology, Linkou Chang Gung Memorial Hospital, Taoyuan City 333, Taiwan

**Keywords:** traditional industries, health promotion behavior, occupational burnout, sleep disturbances

## Abstract

*Background*: Sleep disturbances affect about 40% of the global population and are a common issue among patients seeking medical consultation. There is limited research on sleep disturbances in Taiwan’s traditional industry workforce. *Objective*: This study aims to investigate the correlations between work patterns, health promotion behaviors, occupational burnout, and sleep disturbances among employees in traditional industries. *Methods*: A cross-sectional study was conducted to collect data on the work patterns, health promotion behaviors, occupational burnout, and sleep disturbances within a traditional industry. The study period was from May to June 2023. Data analysis was performed using chi-square tests, independent sample *t*-tests, and logistic regression. *Result*: Data from a total of 365 employees were collected, with 63.3% of the study subjects working in shifts and 47.9% experiencing sleep disturbances. Factors associated with sleep disturbances included personal burnout (OR = 1.07 (95% CI 1.06, 1.09); *p* < 0.001), shift work (OR = 2.67 (95% CI 1.53, 4.65); *p* < 0.001), health responsibility behavior (OR = 0.50 (95% CI 0.33, 0.77); *p* = 0.001), and life appreciation behavior (OR = 0.47 (95% CI 0.29, 0.76); *p* = 0.002). *Conclusion*: Occupational health nurses should regularly assess employees’ sleep status and provide psychological counseling services and health promotion programs to help employees alleviate sleep disturbances.

## 1. Introduction

Sleep disturbances are a common complaint among patients, with insomnia being the most prevalent [1]. Taddei-Allen (2020) [2] reported that annual healthcare costs related to sleep disturbances in the U.S. had reached USD 10 billion. Besides the economic burden, sleep disturbances negatively affect health, impacting social abilities, and work performance, and increase the risks of occupational injuries, accidents, and even suicide [3,4]. They are also linked to higher risks of obesity, type II diabetes, depression, anxiety, cardiovascular disease, and chronic conditions like cancer and dementia [5,6,7]. A study of 22,330 adults from 13 countries found a 36.7% prevalence of sleep disturbances [8]. The use of sleeping pills among Taiwanese people ranks the highest in Asia. According to the National Health Insurance Administration (NHIA) under the Ministry of Health and Welfare, statistics show that outpatient and inpatient populations using sedative-hypnotics continued to increase in 2018. Nationwide, over 4.26 million people use such medications, meaning that on average, at least one in five individuals have taken sleeping pills. The annual prescribed volume exceeds 918 million pills, resulting in significant health insurance expenditure of approximately TWD 2.09 billion [9]. Thus, sleep disturbances remain a significant concern today.

### Background

Traditional industries primarily refer to the manufacturing sector, which involves transforming materials into new products through chemical or physical processes using either human labor or machinery [10,11]. In March 2024, around 2.993 million people, or 25.84% of Taiwan’s labor force, were employed in manufacturing [12]. Known as the “Kingdom of Manufacturing”, Taiwan’s plastics-related industries are a key part of its traditional industrial sector. In 2023, the rubber and plastics machinery industry’s export value reached USD 870 million, making Taiwan the sixth largest exporter globally. This industry is crucial for Taiwan’s economic development and stability.

In Asia, studies on sleep disturbances among traditional industry workers found prevalence rates of 22.6% among electronic product manufacturers in China [13], 35.3% to 42.0% among Chinese steel factory workers [14,15], and 16.7% among Chinese textile workers [16]. Other rates include 4.5% among South Korean manufacturing workers [17], 37.6% among Japanese construction workers [18], and 33.2% in Indian construction workers [19]. These figures show significant variation across countries, but there are no available data on sleep disturbances among workers in Taiwan’s traditional industries. This study aims to address this gap by investigating sleep disturbances in employees within Taiwan’s large plastic products manufacturing sector, providing insights into their prevalence and influencing factors, along with suggested solutions.

Studies have shown that sleep disturbances can lead to increased stress, job insecurity, and poor job performance, contributing to absenteeism [18,20,21]. Recent studies have identified several factors influencing sleep disturbances among employees, including background characteristics such as being female, of older age [22,23,24], smoking, having a lower educational attainment, alcohol use, and chronic illnesses [7,17,19,22,23,24,25]. While these are established risk factors, prior research has primarily focused on non-traditional industry workers. Given that work patterns in traditional industries differ significantly, the factors affecting sleep disturbances in these settings may also vary. Manufacturing remains a major industry in many countries, but previous research on manufacturing employees has largely centered on the health risks associated with occupational exposure [26]. Early identification of the prevalence and contributing factors of sleep disturbances is crucial, as it not only improves employee health and quality of life but also helps to reduce national healthcare costs. Despite its importance, there is a notable lack of studies examining sleep disturbances and their influencing factors among traditional manufacturing industry employees in Taiwan.

Work-related factors linked to sleep disturbances include shift work, increased stress, excessive job involvement, shorter rest periods, and working over 40 h per week [25,27,28,29]. Many industries use shift work to boost productivity, which raises the likelihood of sleep issues [30]. Given that traditional manufacturing employees often work shifts, they may face a higher risk of sleep disturbances. This study seeks to examine the association between work-related factors and sleep disturbances among these workers.

Occupational burnout is a major cause of sleep disturbances. To stay competitive, employers often use flexible labor strategies that lead to workplace fatigue, or burnout, resulting in issues like insomnia, depression, job dissatisfaction, and absenteeism [31,32]. Recognized globally as a crisis, burnout was included in the ICD-11 by the World Health Organization in 2019 [33]. Higher burnout levels increased the risk of sleep disturbances [34]. Studies have mostly focused on professionals like physicians, nurses, teachers, and tech workers, but improving sleep health can alleviate burnout and enhance productivity across sectors [35]. Despite the labor-intensive nature of traditional industries, physical demands do not necessarily lead to better sleep, indicating a need to study the connection between burnout and sleep disturbances in these settings.

Occupational burnout and sleep disturbances are not only concerns in Taiwan but also key policy issues in many countries. According to Vega-Escaño et al. [34], improving sleep disturbances among employees can enhance productivity and reduce occupational burnout. Literature reviews show that studies on sleep disturbances and occupational burnout typically focus on shift workers such as physicians, teachers, police officers, firefighters, nurses, and technology professionals [36,37,38]. However, traditional industry workers also frequently engage in shift work, which, consequently, increases the likelihood of sleep disturbance.

Health promotion involves empowering individuals to improve their health, with key behaviors including health responsibility, proper nutrition, stress management, and exercise [39]. Adult health behaviors (AHBs) like exercise, stress management, and healthy eating are strongly linked to life satisfaction [40]. Regular physical activity is negatively correlated with sleep disturbances, reducing their incidence [41]. Healthy eating also plays a significant role, while insufficient activity and sedentary behavior increase sleep disturbance risks [42]. Sleep disturbances can lower physical energy, reducing engagement in physical activities [43], thus creating a cycle where poor sleep and unhealthy lifestyles exacerbate each other [43,44].

Moreover, research on the relationship between work patterns, health promotion behaviors, occupational burnout, and sleep disturbances in traditional industry workers is even more scarce, warranting attention and further exploration. Therefore, this study aims to investigate the correlation among work patterns, health promotion behaviors, occupational burnout, and sleep disturbances among employees in traditional industries.

## 2. Methods

### 2.1. Research Design

This is a cross-sectional research study that utilizes structured questionnaires to collect data. It aims to explore the relationships between health promotion behaviors, occupational burnout, sleep disturbances, and the related factors of sleep disturbances among employees in traditional industries. The Reporting of Observational Studies in Epidemiology (STROBE) cross-sectional study checklist [45] was used as a reporting guide.

### 2.2. Study Setting

The study setting was a traditional industrial factory in northern Taiwan that primarily engaged in plastic raw material and plastic products manufacturing. There were five plants in this traditional industrial factory.

### 2.3. Participants

Participants in this study were employees of a traditional industrial factory located in northern Taiwan. Inclusion criteria: (1) Currently employed staff with at least three months of experience in traditional industries. (2) Employees aged 18 years or older who provided consent to participate in this study. Exclusion criteria: pregnant individuals.

### 2.4. Sampling and Sample Size

The study participants were selected using purposive sampling, which was conducted in the traditional industrial factory. Following discussions with departmental management regarding safety considerations, the research was confined to plant 1.

The required sample size for correlation analysis was calculated using the G*Power version 3.1.9. software program (Heinrich Heine University Düsseldorf in Düsseldorf, Germany) for logistic regression analysis (“Z-test for Logistic regression”) [46]. Setting the odds ratio (OR) to 1.5, H_0_ to 0.5, significance level (α) to 0.05, and power to 0.9, the minimum sample size needed was 275. Data collection took place from May 2023 to June 2023, with 375 questionnaires distributed and all 365 valid questionnaires returned, yielding a response rate of 97.3%. Participation in this study was voluntary, and employees were told that a small gift (an insulated lunch bag or a multifunctional massager) would be given as a token of appreciation upon completion of the questionnaire.

### 2.5. Measures

Data were collected using structured questionnaires, which included the following scales:

#### 2.5.1. Demographic Attributes

Demographic attributes included basic attributes and work patterns. Based on the literature, the basic attributes of the questionnaire encompassed variables such as gender, age, education level, marital status, smoking habits, and alcohol consumption. Work patterns included shift work, weekly working hours, and managerial position [7,17,19,22,23,24,25].

#### 2.5.2. Health Promotion Behaviors Questionnaire

This was measured using the “Adult Health Behaviors—Short Form” (AHBs) developed by Chen et al. [40]. Participants completed the questionnaire regarding their lifestyle habits over the past year using a four-point Likert scale. The scoring criteria were as follows: 1 point for “Never” (approximately 0–10%), 2 points for “Sometimes” (approximately 11–50%), 3 points for “Often” (approximately 51–70%), and 4 points for “Always” (approximately 71–100%). Higher scores indicated healthier lifestyles. The scale covered six elements: stress management, physical activity, health responsibility, life appreciation, healthy eating, and oral hygiene behaviors. The internal consistency of the AHBs was 0.85. Lower scores on the six subscales and the total score of the SF-AHB were significantly associated with lower life satisfaction. Test–retest reliability for the total AHB was r = 0.91. The validity of the AHBs was confirmed through qualitative content validation by experts in metabolism/endocrinology, cardiology, dentistry, and nursing, with an average Content Validity Index (CVI) of 0.93 [40].

#### 2.5.3. Occupational Burnout Questionnaire

This study utilized the “Chinese Version of the Occupational Burnout Scale” developed by Yeh et al. (2008) [47], which references the Copenhagen Burnout Inventory (CBI) developed by Danish scholars and the Effort–Reward Imbalance Questionnaire (ERI-Q). The questionnaire assesses burnout experienced in the past week through three subscales: (1) “Personal Burnout”: A comprehensive assessment aimed at measuring the participant’s overall sense of burnout, with sources not limited to work but also potentially stemming from family, social interactions, personal health conditions, etc.; (2) “Work Burnout”: Refers to burnout caused by and attributable to work. This subscale is applicable to individuals who are employed; (3) “Over-commitment to Work”: Measures workers’ behavioral tendency to excessively engage in their work. The Cronbach’s alpha for this scale was 0.86 overall (0.92 for men and 0.90 for women). The Cronbach’s alpha for each subscale ranged between 0.85 and 0.87. The scale used a five-point Likert scale with the following scoring criteria: 5 points for “always” (100 points), 4 points for “often” (75 points), 3 points for “sometimes” (50 points), 2 points for “rarely” (25 points), and 1 point for “never” (0 points). Higher scores indicate a greater degree of perceived burnout [46].

#### 2.5.4. Sleep Disturbances Questionnaire

The Chinese version of the eight-question Athens Insomnia Scale (CAIS-8), jointly translated by Jiang et al. (2009) [48], was used to measure the sleep status of the research subjects in the past month by filling in the questionnaire. The subjective feelings of the subjects regarding their nightly sleep were surveyed. The questionnaire included “night-time symptoms” (time to fall asleep, waking up at night, waking up earlier than expected, total sleep time, and total sleep quality) and “daytime symptoms” (daytime mood, daytime physical function, daytime sleepiness) in a total of eight questions. A four-point Likert scale was used, with each question scored from 0 to 3 points, with a total score of 0 to 24 points. A total score of less than 3 points (inclusive) indicates no sleep disturbances; a total score of 4 to 5 indicates suspected sleep disturbances, and a score greater or equal to 6 indicates sleep disturbances. The dependent variables analyzed at the end of this study were “no” (score 0 to 5 points) and “yes” (score ≧ 6 points) as the cut-off points for sleep disturbances [49]. The Cronbach’s alpha for this scale was 0.82 to 0.84, and 0.89 for our study. In our study, after two weeks, the test–retest reliability coefficient for self-reported sleep disturbances was 0.82.

### 2.6. Data Processing and Analysis

Data entry and statistical analysis were conducted using SPSS version 27.0, with an alpha value set to 0.05 (two-tailed test) as the criterion for statistical significance. Basic attributes were analyzed using percentages, frequency distributions, means, and standard deviations to understand the distribution characteristics of the measurements. Inferential statistics included the chi-squared test to compare categorical variables of the study subjects with sleep disturbances (yes or no); an independent *t*-test to examine the differences in continuous variables of the basic attributes, health promotion scale results, and occupational burnout scale results; and logistic regression to analyze the relationship between independent variables and sleep disturbances (yes or no) to identify factors related to sleep disturbances.

Logistic regression analysis was conducted to identify the factors influencing sleep disturbance, incorporating basic attributes, work patterns, health promotion behaviors, and occupational burnout. The analysis was divided into six models: Model 1—the relationship between work patterns and sleep disturbances; Model 2—the relationship between health promotion behaviors and sleep disturbances; Model 3—the relationship between occupational burnout and sleep disturbances; Model 4—the relationship between work patterns, health promotion behaviors, occupational burnout, and sleep disturbances; and Models 5 and 6—the relationship between weekly working hours, health promotion behaviors, occupational burnout, and sleep disturbances with or without shift work.

### 2.7. Ethical Considerations

The Institutional Review Board approved this study under approval number A202305036. Participants were informed about the study through an information sheet, and data were collected using anonymous self-administered questionnaires. Participants were informed of their right to withdraw from the study at any time. Participants received small gifts after completing the questionnaire.

## 3. Results

### 3.1. Descriptive Statistics of Basic Attributes

This study involved a total of 365 participants, with the majority being male (94.5%). The average age was 43.86 years (SD = 11.74). Among the participants, 60% were married, 73.4% reported no smoking habits, including those who had quit smoking, and 74.2% reported no drinking habits, including those who had quit drinking. Most participants had a high school education (52.3%). A significant portion of the participants were shift workers (63.3%), and the majority worked 40 h or less per week (67.4%). Additionally, 86% of the participants held non-supervisory positions (Table 1).

### 3.2. Descriptive Statistical Analysis Results of Health Promotion Behaviors, Occupational Burnout, and Sleep Disturbances

#### 3.2.1. Health Promotion Behaviors

The original scale had a maximum average score of 4 points. The overall average score on the scale was 2.34 points, with a standard deviation of 0.48. The average scores of each sub-dimension, listed from highest to lowest, were as follows: “oral hygiene behavior”, with an average of 2.56 points (SD = 0.69); “life appreciation behavior”, with an average of 2.46 points (SD = 0.68); “healthy eating behavior”, with an average of 2.44 points (SD = 0.58); “health responsibility behavior”, with an average of 2.32 points (SD = 0.78); and “stress management behavior”, with an average of 2.28 points (SD = 0.56), while the lowest was “exercise behavior”, with an average of 2.01 points (SD = 0.75). The scores for the first five sub-dimensions fell between “occasionally” and “frequently,” while exercise behavior was rated as “occasionally”, indicating 2–3 days per week.

#### 3.2.2. Occupational Burnout

The original scale had a maximum average score of 4 points. The overall average score on the scale was 1.96 points, with a standard deviation of 1.09. The total scores for each sub-dimension, listed from highest to lowest, were as follows: “personal burnout”, with an average of 2.30 points (SD = 0.78); “work-related burnout”, with an average of 2.16 points (SD = 0.81); and “overcommitted to work”, with an average of 2.07 points (SD = 0.75). All three sub-dimensions fell between “sometimes” and “often”.

#### 3.2.3. Sleep Disturbances

A total score of less than or equal to 3 indicated no sleep disturbance, accounting for 32.3% of the subjects. A total score of 4 to 5 indicated suspected sleep disturbance, accounting for 19.7%. A total score of 6 or greater indicated the presence of sleep disturbance, with 47.9% of the subjects experiencing sleep disturbances (Table 2).

### 3.3. Bivariable Analysis of Factors Affecting Sleep Disturbance

A bivariable analysis was conducted to understand the relationship between the study subjects’ basic attributes, health promotion behaviors, occupational burnout, and sleep disturbances. The dependent variable was the presence or absence of sleep disturbances, while the independent variables included basic attributes, health promotion behaviors, and occupational burnout. Categorical variables were analyzed using the chi-squared test, and continuous variables were analyzed using the independent *t*-test. The results are as follows: significant differences were found for shift work (*p* < 0.001), weekly working hours (*p* = 0.027), stress management behavior (*p* = 0.001), exercise behavior (*p* < 0.001), health responsibility behavior (*p* < 0.001), life appreciation behavior (*p* < 0.001), healthy diet behavior (*p* < 0.001), oral hygiene behavior (*p* < 0.001), personal burnout (*p* < 0.001), work-related burnout (*p* < 0.001), and overcommitment to work (*p* = 0.001). Among the participants, 48.7% of the males and 35% of the females experienced sleep disturbances. Sleep disturbances were reported by 51.4% of those with a partner and 45.7% of those without, by 51.0% of smokers and 46.8% of non-smokers, by 48.9% of drinkers and 47.6% of non-drinkers, as well as by 48.8% of those with a high school education or lower and 47.1% of those with a college education or higher. However, none of these variables demonstrated a statistically significant correlation (Table 3).

### 3.4. Analysis of Factors Associated with Sleep Disturbances

We included independent variables with significant differences in the bivariate analysis into the logistic regression analysis. For Model 1, logistic regression analysis was conducted using work patterns as independent variables and sleep disturbances as the dependent variable. The results are as follows: shift work (yes/no) (OR = 2.79 (95% CI 1.74, 4.49); *p* < 0.001), weekly working hours (41~48 h/≤40 h) (OR = 2.23 (95% CI 1.37, 3.65); *p* = 0.001), and weekly working hours (˃48 h/≤40 h) (OR = 3.89 (95% CI 1.19, 12.99); *p* = 0.027). These results indicate that higher frequencies of shift work and working more than 40 h per week are associated with a higher likelihood of sleep disturbances. The explanatory power (R^2^) of this model was 0.09 (Model 1; Table 4).

For Model 2, using health promotion behaviors as independent variables and sleep disturbance as the dependent variable, the results are as follows: health responsibility behavior (OR = 0.65 (95% CI 0.45, 0.95); *p* = 0.026), life appreciation behavior (OR = 0.45 (95% CI 0.28, 0.70); *p* < 0.001), and healthy eating behavior (OR = 0.54 (95% CI 0.33, 0.89); *p* = 0.015). These results indicate that lower scores in health responsibility behavior, life appreciation behavior, and healthy eating behavior are associated with a higher likelihood of sleep disturbances. The explanatory power (R^2^) of this model was 0.20 (Model 2; Table 4).

For Model 3, occupational burnout was used as the independent variable and sleep disturbance as the dependent variable. The results are as follows: personal burnout (OR = 1.05 (95% CI 1.03, 1.08); *p* < 0.001) and work-related burnout (OR = 1.03 (95% CI 1.01, 1.05); *p* = 0.01). These results indicate that higher personal and work-related burnout scores are associated with a higher likelihood of sleep disturbances. The explanatory power (R^2^) of this model was 0.32 (Model 3; Table 4).

For Model 4, stepwise logistic regression analysis was conducted using work patterns, health promotion behaviors, and occupational burnout as independent variables and sleep disturbances as the dependent variable. The results are as follows: personal burnout (OR = 1.07 (95% CI 1.06, 1.09); *p* < 0.001) and shift work (Yes/No) (OR = 2.67 (95% CI 1.53, 4.65); *p* = 0.001), health responsibility behavior (OR = 0.50 (95% CI 0.33, 0.77); *p* = 0.001), and life appreciation behavior (OR = 0.47 (95% CI 0.29, 0.76); *p* = 0.002). These results indicate that a higher likelihood of sleep disturbances is associated with higher scores in personal burnout and shift work but lower scores in health responsibility behavior and life appreciation behavior. The explanatory power (R^2^) of this model was 0.45. Data from a total of 365 subjects were collected. In terms of the proportion of sleep disturbances among shift and non-shift workers, the effect size was approximately 0.91 (Model 4; Table 4).

For Model 5, 6 since shift work is related to sleep disturbances, non-shift workers (Model 5: *n* = 231) and shift workers (Model 6: *n* = 134) were separated, and the work patterns were classified accordingly. Work patterns, health promotion behaviors, and occupational burnout were set as independent variables, and sleep disturbance as the dependent variable. A stepwise logistic regression analysis was conducted (to avoid interaction effects) to perform sensitivity analyses to confirm the robustness of the findings. The variables included in the models for both shift and non-shift workers were weekly working hours, health promotion behaviors (stress management behaviors, physical activity behaviors, health responsibility behaviors, life appreciation behaviors, healthy eating behaviors, oral hygiene behaviors) and occupational burnout (personal burnout, work-related burnout, overcommitment to work). For non-shift work, the results are as follows: personal burnout (OR = 1.09 (95% CI 1.06, 1.12); *p* < 0.001) and health responsibility (OR = 0.35 (95% CI 0.21, 0.57); *p* < 0.001); the explanatory power (R^2^) of this model was 0.44. Not applicable were weekly working hours, health promotion behaviors (stress management behaviors, physical activity behaviors, life appreciation behaviors, healthy eating behaviors, oral hygiene behaviors, and occupational burnout (work-related burnout, overcommitment to work) (Model 5; Table 4)

For shift work, the results are as follows: work-related burnout (OR = 1.07 (95% CI 1.04,1.10); *p* < 0.001), life appreciation behavior (OR = 0.27 (95% CI 0.11, 0.66); *p* = 0.004), and stress management (OR = 0.32 (95% CI 0.11, 0.93); *p* = 0.035); the explanatory power (R^2^) of this model was 0.47. Not applicable were weekly working hours, health promotion behaviors (physical activity behaviors, health responsibility behaviors, healthy eating behaviors, oral hygiene behaviors) and occupational burnout (personal burnout, overcommitment to work) (Model 6; Table 4).

## 4. Discussion

Our study found that shift work, health responsibility, life appreciation, and personal burnout are associated with sleep disturbances. For shift workers, work-related burnout, life appreciation, and stress management are linked to sleep disturbances, while for non-shift workers, personal burnout and health responsibility are associated with sleep disturbances.

This study highlights that sleep disturbances are more prevalent in the manufacturing sector under investigation compared with other Asian manufacturing industries [13,14,15,16,17,18]. The elevated rates observed in this study may stem from the high proportion of shift workers, the physically demanding nature of their tasks, and the predominantly male workforce. Shift work disrupts circadian rhythms, potentially leading to prolonged sleep deprivation, which can obscure early signs of productivity decline until the symptoms become severe.

Cognitive behavioral therapy (CBT), including internet-based CBT, has been identified as an effective intervention for managing sleep disturbances, offering a practical solution for individuals unable to access in-person therapy. Employers are encouraged to recognize the detrimental effects of sleep disturbances on productivity and to consider workplace health initiatives. These could include sleep health programs, subsidies for supportive technology, and educational campaigns to promote healthy sleep hygiene, all of which can contribute to improved sleep quality and overall well-being [50,51].

The findings also emphasize the notable difference in sleep disturbances between shift and non-shift workers [52,53]. Shift workers are more susceptible to disrupted sleep due to their irregular schedules and physical strain, highlighting the need for targeted interventions tailored to their unique challenges [54]. Shift work is essential in modern industries due to economic demands but often leads to natural sleep–wake cycle disturbances, causing difficulties in sleep and daytime alertness. Treatments include lifestyle changes, light therapy, and medication. Recommendations include adjusting to a two-shift system to stabilize schedules and increasing exposure to bright light for night-shift workers, which helps regulate circadian rhythms. Morning light exposure reduces melatonin, aiding in earlier sleep times, while night shifts benefit from moderate-intensity light to reduce sleepiness [55,56]. Therefore, it is recommended to utilize bright light in night-shift work environments to improve sleep disturbances among shift workers.

Our findings are consistent with prior research conducted among workers in an automobile parts manufacturing company [55], which demonstrated that reducing working hours per shift and minimizing night shifts can significantly improve sleep health. These results underscore the importance of implementing similar interventions in traditional industrial settings, such as optimizing shift schedules to reduce the physical and mental strain on employees. For instance, policies aimed at limiting prolonged night shifts or promoting shorter shifts could be effective strategies to mitigate sleep disturbances and to enhance overall well-being.

Sleep is one of the three pillars of health, alongside nutrition and exercise, and is seen as a modifiable goal for an individual’s well-being. To address rising healthcare costs, innovative employers are investing in their workforce through Worksite Health Promotion programs [57]. Our study highlights the critical role of health promotion behaviors, including life appreciation, healthy eating, and health responsibility, in mitigating sleep disturbances. Employees who maintain a positive mindset, such as practicing daily gratitude and optimism, report better sleep quality [58]. Additionally, adopting healthier eating habits, such as a Mediterranean diet rich in whole grains, healthy fats, and lean proteins, can further improve sleep patterns and reduce the risk of sleep disturbances [42,59,60]. Proactive health management, reflected in higher health responsibility scores, also enhances self-regulation and overall well-being. These findings emphasize the need for workplace interventions, such as nutrition counseling and wellness programs, to foster these behaviors and to promote better sleep health among employees [61].

Although previous research by Xie et al. (2021) [62] and Bjornsdottir et al. (2024) [63] indicates that regular exercise improves sleep quality and reduces insomnia, this study did not find a clear link, possibly because most participants did not engage in regular exercise. Encouraging regular physical activity, especially for shift workers, could improve sleep quality [64,65]. Future initiatives might include smart bands, stair-climbing campaigns, and accessible workplace exercise options, with follow-up studies to confirm exercise’s benefits for sleep disturbances. In Table 3, both groups with and without sleep disturbances showed low exercise scores. Future studies could explore the potential of encouraging regular physical activity to examine its relationship with sleep disturbances [66].

Higher personal and work burnout scores correlate with increased sleep disturbances. A study on 18,744 Korean workers confirmed that workplace burnout raises the likelihood of sleep disturbances [33], and similar correlations were found among Hungarian postal workers [67]. Research on physicians [68] and nurses [69] also supports this link, aligning with the findings of this study.

Contrary to the belief that being physically tired promotes better sleep, labor-intensive work does not necessarily improve sleep quality. Workers often endure long hours of standing and heavy lifting, which can increase physical strain without reducing burnout. Providing a comfortable work environment and short breaks (20–30 min) can help lower burnout and improve sleep, especially for night-shift workers [70,71].

Meditation practices, such as transcendental meditation and Sudarshan Kriya yoga, have been shown to effectively reduce stress, burnout, and insomnia [72,73,74]. Encouraging leisure activities and hosting workshops on mental and physical health, including aromatherapy, muscle relaxation, and deep breathing, can also alleviate stress, reduce burnout, and improve sleep quality.

Addressing burnout requires a comprehensive approach that includes effective workplace resource management and opportunities for skill development. At the employer level, establishing conditions that foster optimal work performance is essential. Providing emotional support and resilience training equips employees with valuable resources to manage stress. To improve job performance, optimizing workloads, reducing excessive working hours, and promoting task sharing are critical strategies. The integration of automation and artificial intelligence (AI) technologies further enhances work processes, reduces employee burnout, and boosts efficiency. Moreover, AI robots can be utilized to handle repetitive or demanding tasks, alleviating the burden on individuals and mitigating personal burnout. On a personal level, self-care is indispensable. Prioritizing physical fitness, ensuring adequate rest, and practicing effective stress management are vital to maintaining overall well-being and preventing burnout. Through these combined efforts, both organizations and individuals can create a healthier and more productive work environment [75].

Our study found that shift workers’ work-related burnout, life appreciation behavior, and stress management are linked to sleep disturbances, while for non-shift workers, personal burnout and health responsibility are associated with sleep disturbances. Possible explanations include the fact that shift workers face irregular work hours, disrupting their circadian rhythm, increasing physical and mental stress, and affecting sleep quality. Work-related burnout may reflect resource depletion due to long or night shifts. Non-shift workers, with more stable work hours, face pressures from balancing work and family responsibilities, leading to personal burnout, which affects their health responsibility and sleep. Non-shift workers’ burnout often stems from the dual pressures of work and family, such as caring for children and aging parents [76]. This can hinder their ability to manage health, affecting sleep. In contrast, shift workers’ primary challenges are irregular work and lifestyle, with less family-related pressures.

These differences highlight the need for tailored health promotion strategies. For shift workers, interventions should focus on stress management and life appreciation; for non-shift workers, strategies should address personal burnout and enhance health responsibility. Future research should explore the role of family and social factors in sleep disturbances.

## 5. Conclusions

This study explored the factors associated with sleep disturbances among employees in traditional industries, with a particular focus on differences between shift and non-shift workers. The findings indicate that sleep disturbances are influenced by shift work and specific health promotion behaviors, including health responsibility, life appreciation, and healthy eating, as well as personal and work-related burnout. For non-shift workers, personal burnout and health responsibility are key factors affecting sleep quality. In contrast, for shift workers, work-related burnout, life appreciation behaviors, and stress management have a more significant impact on sleep quality. These findings highlight the unique challenges faced by each group and underscore the importance of tailored interventions to address their specific needs.

### Limitations and Recommendations

This study has three limitations. First, sleep disturbances were measured subjectively using self-reported questionnaires, without employing objective wearable devices to monitor sleep. Second, data were collected from a single site within a traditional factory, which limits the generalizability of the findings. Third, as a cross-sectional study, it only allows for the exploration of associations and does not enable the investigation of causal relationships.

Future studies should integrate objective methods, such as wearable devices or polysomnography, to monitor sleep patterns and disturbances. These tools can offer more precise and comprehensive data, minimizing potential biases linked to self-reported measures. Additionally, to improve the generalizability of findings, future research should broaden the data collection to include multiple sites across various traditional industries. This would facilitate comparisons across different work environments and populations. Moreover, longitudinal studies are recommended to establish causal relationships among health promotion behaviors, occupational burnout, and sleep disturbances. By tracking changes over time, these studies can provide deeper insights into the dynamic interactions between these variables.

## Figures and Tables

**Table 1 healthcare-13-00051-t001:** Descriptive statistics of subjects’ basic attributes and work patterns (*n* = 365).

Characteristics	*n*	(%)	M	SD
Basic attributes				
Age (years)			43.86	11.74
Gender				
Male	345	94.5		
Female	20	5.50		
Marital Status				
Single	146	40.0		
Married	219	60.0		
Smoking Habits				
None	269	73.4		
Current Smoker	96	26.3		
Drinking Habits				
None	271	74.2		
Current drinker	94	25.8		
Education level				
Senior high or below	191	52.3		
Associate degree or above	174	47.7		
Work patterns				
Shift Work				
No	134	36.7		
Yes	231	63.3		
Weekly Work (hours)				
≦40	246	67.4		
41~48	106	29.0		
>48	13	3.60		
Managerial position				
No	314	86.0		
Yes	51	14.0		

Note: M  =  mean; SD  =  standard deviation.

**Table 2 healthcare-13-00051-t002:** Descriptive statistics of health promotion behaviors, occupational burnout, and sleep disturbances (*n* = 365).

Characteristics	*n*	(%)	M	SD
**Health promotion behaviors**			2.34	0.48
Stress management behaviors			2.28	0.56
Physical activity behaviors			2.01	0.75
Health responsibility behaviors			2.32	0.78
Life appreciation behaviors			2.46	0.68
Healthy eating behaviors			2.44	0.58
Oral hygiene behaviors			2.56	0.69
**Occupational burnout**			1.96	1.09
Personal burnout			2.30	0.78
Work-related burnout			2.16	0.81
Overcommitment to work			2.07	0.75
**Sleep condition**				
No sleep disturbance	118	32.3		
Suspected sleep disturbance	72	19.7		
Sleep disturbance	175	47.9		

**Table 3 healthcare-13-00051-t003:** Analysis of differences in basic attributes, health promotion behaviors, occupational burnout, and sleep disturbances (*n* = 365).

Characteristics	*n*	Sleep Disturbances			
		Yes *n* (%)	No *n* (%)	*χ^2^*/*t*	*p*
Basic attributes					
Age	365	43.3 ± 11.5	44.3 ± 12.0	0.17	0.42
Gender				1.42	0.23
Male	345	168 (48.7)	177 (51.3)		
Female	20	7 (35.0)	13 (65.0)		
Marital Status				1.14	0.29
Single	146	75 (51.4)	71 (48.6)		
Married	219	100 (45.7)	119 (54.3)		
Smoking Habits				0.50	0.55
None	269	126 (46.8)	143 (53.2)		
Current Smoker	96	49 (51.0)	47 (49.0)		
Drinking Habits				0.05	0.91
No	271	129 (47.6)	142 (52.4)		
Yes	94	46 (48.9)	48 (51.1)		
Education level				0.89	0.83
Senior high or below	191	93 (48.8)	98 (51.3)		
Associate degree or above	174	82 (47.1)	92 (52.9)		
Work patterns					
Shift Work				12.47	<0.001
No	134	48 (35.8)	86 (64.2)		
Yes	231	127 (55.0)	104 (45.0)		
Weekly Work (hours)				7.20	0.027
≦40	246	106 (43.1)	140 (56.9)		
41~48	106	61 (57.5)	45 (42.5)		
˃48	13	8 (61.5)	5 (38.5)		
Managerial position				0.19	0.66
No	314	152 (48.4)	162 (51.6)		
Yes	51	23 (45.1)	28 (54.9)		
Health promotion behaviors					
Stress management behaviors	365	2.18 ± 0.46	2.37 ± 0.63	3.31	0.001
Physical activity behaviors	365	1.87 ± 0.62	2.14 ± 0.84	3.55	<0.001
Health responsibility behaviors	365	2.09 ± 0.67	2.53 ± 0.82	5.68	<0.001
Life appreciation behaviors	365	2.22 ± 0.60	2.69 ± 0.68	6.96	<0.001
Healthy eating behaviors	365	2.26 ± 0.54	2.61 ± 0.58	5.60	<0.001
Oral hygiene behaviors	365	2.43 ± 0.64	2.70 ± 0.72	3.82	<0.001
Occupational burnout					
Personal burnout	365	2.69 ± 0.78	1.95 ± 0.59	–10.10	<0.001
Work-related burnout	365	2.53 ± 0.84	1.83 ± 0.62	–8.94	<0.001
Overcommitment to work	365	2.20 ± 0.78	1.95 ± 0.71	–3.24	0.001

**Table 4 healthcare-13-00051-t004:** Logistic regression analysis of work patterns, health promotion behaviors, occupational burnout, and sleep disturbances (*n* = 365).

Characteristics	Model 1 OR (95% CI)	Model 2 OR (95% CI)	Model 3 OR (95% CI)	Model 4 OR (95% CI)	Model 5 OR (95% CI)	Model 6 OR (95% CI)
**Work patterns**						
Shift Work (Yes/No ^#^)	2.79 (1.74, 4.49) ***			2.67 (1.53, 4.65) **		
Weekly Work (hours)						
41~48/≦40 ^#^	2.23 (1.37, 3.65) **			N/A	N/A	N/A
˃48/≦40 ^#^	3.89 (1.19, 12.99) *			N/A	N/A	N/A
**Health promotion behaviors**						
Stress management		1.16 (0.72, 1.87)		N/A	N/A	0.32 (0.11, 0.93) *
Physical activity		1.01 (0.71, 1.44)		N/A	N/A	N/A
Health responsibility		0.65 (0.45, 0.95) *		0.50 (0.33, 0.77) **	0.35 (0.21, 0.57) ***	N/A
Life appreciation		0.45 (0.28, 0.70) ***		0.47 (0.29, 0.76) **	N/A	0.27 (0.11, 0.66) **
Healthy eating		0.54 (0.33, 0.89) *		N/A	N/A	N/A
Oral hygiene		1.14 (0.77, 1.70)		N/A	N/A	N/A
**Occupational burnout**						
Personal burnout			1.05 (1.03, 1.08) ***	1.07 (1.06, 1.09) ***	1.09 (1.06, 1.12) ***	N/A
Work-related burnout			1.03 (1.01, 1.05) **	N/A	N/A	1.07 (1.04, 1.10) ***
Overcommitment to work			0.99 (0.98, 1.01)	N/A	N/A	N/A
R^2^	0.09	0.20	0.32	0.45	0.44	0.47

Note: * *p* < 0.05; ** *p* < 0.01; *** *p* < 0.001; ^#^ reference group. OR = Odds ratio. CI = Confidence Interval. N/A = Not Applicable.

## Data Availability

The data presented in this study are available on request from the corresponding author.

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
