# Peer review of "Exploring Health Promotion Behaviors, Occupational Burnout, and Sleep Disturbances in Traditional Industry Workers"

_healthcare, 2024, doi:10.3390/healthcare13010051_

Round 1

Reviewer 1 Report

Comments and Suggestions for Authors

Dear authors and the editor,

This manuscript addresses an important topic by exploring the relationships between health promotion behaviors, occupational burnout, and sleep disturbances among workers in traditional industries. The study's scope is significant, as it focuses on a population often overlooked in sleep research. While the manuscript is well-structured and provides valuable insights, there are several areas requiring refinement to enhance clarity, methodological rigor, and interpretability. Below is a detailed review of each section.

Introduction

The introduction effectively establishes the significance of sleep disturbances and their impact on worker health and productivity. It provides a broad background with relevant global and regional data. However, the literature presented should be more focused, the  relevance is unevenly maintained, as some references (e.g., global statistics on sleep disturbances) lack direct applicability to Taiwanese traditional industries. While the gap in local research is noted, the cited studies predominantly focus on other regions or occupational settings, limiting their contextual transferability. More targeted literature emphasizing the Taiwanese workforce or similar industrial settings in Asia would enhance credibility and specificity. The connection between prior research and the study's aim is not articulated clearly.

Methods

The study assumes that the work environments and demographics (e.g., industry type (textile, food etc), labor intensity, shift work patterns) of all five factories are similar. We need more information and concrete data about these factories. If possible, report data on all five factories to identify any significant differences in demographics, working conditions, or outcomes like sleep disturbances and burnout.

The workforce size for each factory was not specified in the study, but this is critical for understanding the representativeness. Total workforce in Plant 1in factory 1, 2,3,4,5 and the numbers recruited for each factory in the sample

Explain why Plant 1 was chosen and how its characteristics align with or differ from the other factories Provide details on how representative Plant 1 is of the broader population

The phrasing "different factories, same Plant 1" is contradictory and potentially confusing, as the terms "factory" and "plant" are often used interchangeably to describe the same concept: a single manufacturing facility

Results

Readers need to understand the specific research questions each model aims to address. For instance, does Model 1 establish baseline relationships, and does Model 4 test interactions or cumulative effects?

Justify why certain independent variables were selected or excluded in each model. For example:

Why were "shift work" and "weekly working hours" prioritized in early models?

Why were health promotion behaviors and burnout introduced in subsequent models?

The increasing R² values (e.g., from Model 1 to Model 4) indicate stronger predictive ability, but it is important to explain why the added variables improve the model.

While long working hours and burnout are significant predictors, the interplay between these factors (e.g., whether burnout mediates the relationship between long hours and sleep disturbances) is unexplored. interactions or mediation analyses to explore how variables like burnout and working hours interrelate.

Tables are dense and could be simplified for better readability.

Discussion

The flow from statistical findings to workplace interventions is abrupt. For example, the jump from odds ratios for shift work to the recommendation for two-shift systems lacks intermediary discussion on the mechanisms through which shift work disrupts sleep.

The relationship between "health responsibility" and reduced sleep disturbances is discussed, but the pathways (e.g., better self-care or reduced burnout) remain speculative. This weakens the logical connection.

While workplace interventions are proposed (e.g., shift adjustments, health promotion programs), these should be directly tied to the statistical findings, with clearer prioritization based on effect sizes or population needs

The recommendation’s focus is on the wellness interventions in the study but should be more on working conditions.

Strengthen the discussion of non-significant results and propose hypotheses for future research. Discuss how local cultural and regulatory contexts influence findings and their applications.

Limitations

No detailed information is given about differences between the factories. The methodology and results assume that the five factories are similar in terms of work environment, workforce composition, and processes. However, this assumption is not explicitly substantiated, and any differences between factories are not explored or discussed. This is a limitation that could impact the study's generalizability and validity. If the factories belong to different sectors (e.g., textiles, rubber manufacturing, plastics), their work environments, health risks, and schedules could vary significantly. Shift rotation policies, average working hours, and job demands could differ. Geographic differences in the locations of the factories (e.g., urban vs. rural) might affect employees’ commute times, access to healthcare, or living conditions

Author Response

Reviewer1

Comments 1: [The introduction effectively establishes the significance of sleep disturbances and their impact on worker health and productivity. It provides a broad background with relevant global and regional data. However, the literature presented should be more focused, the relevance is unevenly maintained, as some references (e.g., global statistics on sleep disturbances) lack direct applicability to Taiwanese traditional industries. While the gap in local research is noted, the cited studies predominantly focus on other regions or occupational settings, limiting their contextual transferability. More targeted literature emphasizing the Taiwanese workforce or similar industrial settings in Asia would enhance credibility and specificity. The connection between prior research and the study's aim is not articulated clearly.

Response 1-1: Thank you for pointing this out. We agree with this comment. We have added text in the manuscript, please see page 1, lines 37-43.

[The use of sleeping pills among Taiwanese people ranks first in Asia. According to the National Health Insurance Administration (NHIA) under the Ministry of Health and Welfare, statistics show that the outpatient and inpatient population using sedative-hypnotics continued to increase in 2018. Nationwide, over 4.26 million people used such medications, meaning that on average, at least one in five individuals has taken sleeping pills. The annual prescribed volume exceeds 918 million pills, resulting in significant health insurance expenditures of approximately NT$2.09 billion.]

Response 1-2: Thank you for pointing this out. We agree with this comment. We have revised text in the manuscript, please see page 2, lines 89-95.

[Occupational burnout and sleep disturbances are not only concerns in Taiwan but also key policy issues in many countries. According to Vega-Escaño et al. [37], improving sleep disturbances among employees can enhance productivity and reduce occupational burnout. Literature reviews show that studies on sleep disturbances and occupational burnout typically focus on shift workers such as physicians, teachers, police officers, firefighters, nurses, and technology professionals [38-41]. However, traditional industry workers also frequently engage in shift work, which consequently increases the likelihood of sleep disturbance].

Comments 2: [Methods]

The study assumes that the work environments and demographics (e.g., industry type (textile, food etc.), labor intensity, shift work patterns) of all five factories are similar. We need more information and concrete data about these factories. If possible, report data on all five factories to identify any significant differences in demographics, working conditions, or outcomes like sleep disturbances and burnout.

The workforce size for each factory was not specified in the study, but this is critical for understanding the representativeness. Total workforce in Plant 1in factory 1, 2,3,4,5 and the numbers recruited for each factory in the sample

Explain why Plant 1 was chosen and how its characteristics align with or differ from the other factories Provide details on how representative Plant 1 is of the broader population

The phrasing "different factories, same Plant 1" is contradictory and potentially confusing, as the terms "factory" and "plant" are often used interchangeably to describe the same concept: a single manufacturing facility

Response 2: Thank you for pointing this out. We appreciate this insightful comment.

  1. The number of participants in Plant 1 met the minimum required sample size.
  2. The researchers used a chi-square test to analyze the average age of the all population, which was 44.5 years (t = -1.03, p = 0.3), indicating no significant difference between the average age of the participants in Plant 1 and the all population.
  3. The unit supervisor of Plant 1 agreed to proceed with the recruitment and has signed the project consent form.

Comments 3: Results

Readers need to understand the specific research questions each model aims to address. For instance, does Model 1 establish baseline relationships, and does Model 4 test interactions or cumulative effects?

Justify why certain independent variables were selected or excluded in each model. For example:

Why were "shift work" and "weekly working hours" prioritized in early models?

Why were health promotion behaviors and burnout introduced in subsequent models?

The increasing R² values (e.g., from Model 1 to Model 4) indicate stronger predictive ability, but it is important to explain why the added variables improve the model.

While long working hours and burnout are significant predictors, the interplay between these factors (e.g., whether burnout mediates the relationship between long hours and sleep disturbances) is unexplored. interactions or mediation analyses to explore how variables like burnout and working hours interrelate.

Tables are dense and could be simplified for better readability.

Response 3: Thank you for pointing this out. We agree with your comment and have made the necessary revisions to address it. The statistics have been re-analyzed, and the results have been updated accordingly. To avoid interaction effects, stepwise regression was used in Model 4. The changes can be found in the following sections of the revised manuscript:

**Page 4, lines 211–217 (Table 1):** Updates reflect the revised statistical outcomes and accompanying explanations. 

Revised Text in the Manuscript:

.1. Descriptive statistics of basic attributes

This study involved a total of 365 participants, with the majority being male (94.5%). The average age was 43.86 years (SD = 11.74). Among the participants, 60% were married, 73.4% reported no smoking habits, including those who had quit smoking, and 74.2% reported no drinking habits, including those who had quit drinking. Most participants had a high school education (52.3%). A significant portion of the participants were shift workers (63.3%), and the majority worked 40 hours or less per week (67.4%). Additionally, 86% of the participants held non-supervisory positions. Detailed information is provided in Table 1.

**Page 7, lines 269–277 (Table 4):** Revised data and analysis have been incorporated to ensure clarity and accuracy.

Revised Text in the Manuscript:

[We included the independent variables with significant differences in the bivariate analysis into the logistic regression. For Model 1, logistic regression analysis was conducted using work patterns as independent variables and sleep disturbances as the dependent variable. The results are as follows: shift work (yes/no), OR = 2.79 (95% CI 1.74,4.49), p < 0.001; weekly working hours (41~48 hours/≤40 hours), OR = 2.23 (95% CI 1.37,3.65), p = 0.001; and weekly working hours (˃48 hours/≤40 hours), OR = 3.89 (95% CI 1.19,12.99), p = 0.027. These results indicate that higher frequencies of shift work and working more than 40 hours per week are associated with a higher likelihood of sleep disturbance. The explanatory power (R²) of this model was 0.09. See Table 4 for detailed results.]

**Page 8, lines 285–308.:** This section now includes the updated results and detailed explanations reflecting the re-run statistical analyses.] 

Revised Text in the Manuscript:

[For Model 2, using health promotion behaviors as independent variables and sleep disturbance as the dependent variable, the results are as follows: health responsibility behavior, OR = 0.65 (95% CI 0.45,0.95), p = 0.026; life appreciation behavior, OR = 0.45 (95% CI 0.28,0.70), p < 0.001; and healthy eating behavior, OR = 0.54 (95% CI 0.33,0.89), p = 0.015. These results indicate that lower scores in health responsibility behavior, life appreciation behavior, and healthy eating behavior are associated with a higher likelihood of sleep disturbances. The explanatory power (R²) of this model was 0.20. See Table 4 for detailed results.

For Model 3, using occupational burnout as independent variables and sleep disturbances as the dependent variable. The results are as follows: personal burnout, OR = 1.05 (95% CI 1.03,1.08), p < 0.001, and work-related burnout, OR = 1.03 (95% CI 1.01,1.05), p = 0.01. These results indicate that higher personal and work-related burnout scores are associated with a higher likelihood of sleep disturbances. The explanatory power (R²) of this model was 0.32. See Table 4 for detailed results.

For Model 4, Stepwise logistic regression analysis was conducted using work patterns, health promotion behaviors, and occupational burnout as independent variables and sleep disturbances as the dependent variable. The results are as follows: personal burnout, OR = 1.07 (95% CI 1.06,1.09), p < 0.001 and shift work (Yes/No), OR = 2.67 (95% CI 1.53,4.65), p = 0.001; and. The results are as follows: health responsibility behavior, OR = 0.50 (95% CI 0.33,0.77), p = 0.001; life appreciation behavior, OR = 0.47 (95% CI 0.29,0.76), p = 0.002. These results indicate a higher likelihood of sleep disturbances associated with higher scores in personal burnout, shift work, lower scores in health responsibility behavior and life appreciation behavior. The explanatory power (R²) of this model was 0.45. See Table 4 for detailed results. A total of 365 subjects were collected. In terms of the proportion of sleep disturbances among shift and non-shift workers, the effect size is approximately 0.91.]

Comments 4: Discussion

The flow from statistical findings to workplace interventions is abrupt. For example, the jump from odds ratios for shift work to the recommendation for two-shift systems lacks intermediary discussion on the mechanisms through which shift work disrupts sleep. 4.2

The relationship between "health responsibility" and reduced sleep disturbances is discussed, but the pathways (e.g., better self-care or reduced burnout) remain speculative. This weakens the logical connection.

While workplace interventions are proposed (e.g., shift adjustments, health promotion programs), these should be directly tied to the statistical findings, with clearer prioritization based on effect sizes or population needs 4.3

The recommendation’s focus is on the wellness interventions in the study but should be more on working conditions.

Response 4: Thank you for pointing this out. We agree with this comment. We have revised text in the manuscript, please see page 9, lines 348-357, lines 360-361, lines 388-391, page 10, lines 406-417.

Revised Text in the Manuscript:

[A study, conducted among workers in an automobile parts manufacturing company, utilized the Difference-in-Difference (DID) method to examine changes in sleep duration and the prevalence of poor sleep quality among shift and non-shift workers before and after adjustments to the work schedule. In 2010, the company transitioned from a weekly rotating continuous two-shift system to a weekly rotating discontinuous two-shift system, reducing working hours per shift while maintaining two groups (day and night shifts). Despite these adjustments, the weekly rotating shift schedule and the number of workdays remained unchanged. Statistically significant improvements were observed in the experimental group during night shifts: daily sleep duration increased by +0.5 hours, the wake-after-sleep-onset rate decreased by -13.9%, and self-reported poor sleep quality reduced by -34.9% Quitting overnight work led to improved sleep health for shift workers]

[People who prioritize health and nutrition tend to sleep better than those who completely neglect it, consistent with the findings of this study]

Enhance the discussion on non-significant results and propose hypotheses for future research. Consider how local culture and regulatory environments may influence the study outcomes and their applicability. [In Table 3, both groups with and without sleep disturbances showed low exercise scores. Future studies could explore the potential of encouraging regular physical activity to examine its relationship with sleep disturbances]

[Addressing burnout requires a comprehensive approach that includes effective workplace resource management and opportunities for skill development. At the employer level, establishing conditions that foster optimal work performance is essential. Providing emotional support and resilience training equips employees with valuable resources to manage stress. To improve job performance, optimizing workloads, reducing excessive working hours, and promoting task sharing are critical strategies. The integration of automation and artificial intelligence (AI) technologies further enhances work processes, reduces employee burnout, and boosts efficiency. Moreover, AI robots can be utilized to handle repetitive or demanding tasks, alleviating the burden on individuals and mitigating personal burnout. On a personal level, self-care is indispensable. Prioritizing physical fitness, ensuring adequate rest, and practicing effective stress management are vital to maintaining overall well-being and preventing burnout. Through these combined efforts, both organizations and individuals can create a healthier and more productive work environment]

Comments 5: Limitations

No detailed information is given about differences between the factories. The methodology and results assume that the five factories are similar in terms of work environment, workforce composition, and processes. However, this assumption is not explicitly substantiated, and any differences between factories are not explored or discussed. This is a limitation that could impact the study's generalizability and validity. If the factories belong to different sectors (e.g., textiles, rubber manufacturing, plastics), their work environments, health risks, and schedules could vary significantly. Shift rotation policies, average working hours, and job demands could differ. Geographic differences in the locations of the factories (e.g., urban vs. rural) might affect employees’ commute times, access to healthcare, or living conditions

Response 5: Thank you for pointing this out. We agree with this comment. We conducted case recruitment in only one plant within a factory. Although the average age did not significantly differ from that of the parent population, inferences should still be made cautiously. Future studies could consider conducting a census of the entire factory or extending the scope to other traditional industries to enhance the generalizability of the research findings.

We have revised text in the manuscript, page 10, lines 442-455.

Revised Text in the Manuscript:

[Limitations and Recommendations: This study has three limitations. First, sleep disturbances were measured subjectively using self-reported questionnaires, without employing objective wearable devices to monitor sleep. Second, data were collected from a single site within a traditional factory, which limits the generalizability of the findings. Third, as a cross-sectional study, it only allows for the exploration of associations and does not enable the investigation of causal relationships.

Future studies should integrate objective methods, such as wearable devices or poly-somnography, to monitor sleep patterns and disturbances. These tools can offer more precise and comprehensive data, minimizing potential biases linked to self-reported measures. Additionally, to improve the generalizability of findings, future research should broaden the data collection to include multiple sites across various traditional industries. This would facilitate comparisons across different work environments and populations. Moreover, longitudinal studies are recommended to establish causal relationships among health promotion behaviors, occupational burnout, and sleep disturbances. By tracking changes over time, these studies can provide deeper insights into the dynamic interactions between these variables.]

Reviewer 2 Report

Comments and Suggestions for Authors

Thank you for having me as a reviewer for this article - here are my suggestions:

Strengths:

  • The study employs a clear and appropriate cross-sectional design to explore the associations between health promotion behaviors, occupational burnout, and sleep disturbances.
  • The use of validated instruments, such as the Athens Insomnia Scale (CAIS-8) and the Chinese Occupational Burnout Scale, adds credibility.
  • Ethical approval and informed consent were adequately addressed.

Weaknesses:

  • Subjectivity in Data Collection: The reliance on self-reported questionnaires introduces biases such as social desirability or recall bias, potentially affecting the accuracy of the results.
  • Exclusion Criteria: The exclusion criteria (e.g., pregnant participants) are not adequately justified, potentially overlooking a significant subset of the population that might experience different patterns of sleep disturbances.
  • Sample Representation: The study only considers employees from a specific sector in Northern Taiwan, which may limit the generalizability of findings to other industries or regions.
  • Measurement Validity: While validated scales were used, the absence of objective sleep measurements (e.g., actigraphy) reduces the reliability of sleep disturbance data.
  • Cluster Sampling: Random selection from one factory could lead to homogeneity in the sample, reducing variability and potentially biasing results.

Data Analysis

Strengths:

  • The use of logistic regression models is appropriate for identifying factors associated with sleep disturbances.
  • Multiple models (basic attributes, health promotion behaviors, occupational burnout, and combined) strengthen the robustness of the analysis.

Weaknesses:

  • Insufficient Discussion on Multicollinearity: The analysis does not explicitly address potential multicollinearity between independent variables (e.g., burnout and health promotion behaviors), which could distort regression results.
  • Overemphasis on Statistical Significance: While p-values are reported, the study lacks a detailed discussion on effect sizes or the clinical relevance of the findings.
  • Inconsistencies in Reporting: Some analysis outcomes are not sufficiently detailed. For instance, subgroup analyses for gender or specific job roles (administrative vs. shift workers) could provide additional insights.

Conclusion

Strengths:

  • The conclusions align with the study's objectives, emphasizing the role of work-related factors and burnout in sleep disturbances.
  • Practical recommendations for occupational health nurses and employers are well-articulated.

Weaknesses:

  • Overgeneralization: Conclusions suggest a causal relationship between variables, despite the cross-sectional design, which cannot establish causality.
  • Limited Depth in Recommendations: While workplace interventions are suggested, specific implementation strategies or evidence from similar successful programs are not provided.
  • Inconsistent Emphasis: The conclusion downplays certain significant findings, such as the role of shift work and long working hours, which should be given more prominence considering their strong statistical associations.

Language

Strengths:

  • The paper is well-organized, with clear headings and a logical flow between sections.
  • Technical terms are appropriately used, and references are cited correctly.

Weaknesses:

  • Repetitive Language: Some sections (e.g., discussion and results) repeat the same points without adding new insights.
  • Grammatical Errors: There are occasional grammatical mistakes, such as "increases risks of occupational injuries" (should be "increased risks").
  • Awkward Phrasing: Sentences like “employees’ burnout and sleep disturbances are mutual risk factors” could be rephrased for better readability.
  • Overuse of Jargon: Simplifying technical terms or providing definitions for less familiar terms (e.g., "circadian rhythm disorders") could improve accessibility.

Recommendations for Improvement

  1. Methodology:

    • Include objective measures for sleep disturbances, such as actigraphy or polysomnography.
    • Broaden inclusion criteria and ensure diversity in the sample population.
    • Justify the sampling method more explicitly to address potential biases.
  2. Data Analysis:

    • Address potential multicollinearity and include a discussion of effect sizes.
    • Perform subgroup analyses for demographic variables and report findings comprehensively.
    • Use sensitivity analyses to confirm the robustness of findings.
  3. Conclusions:

    • Avoid causal language and clarify that findings indicate associations, not causality.
    • Expand on practical recommendations with specific examples of workplace health programs.
    • Highlight limitations more transparently, especially those related to the cross-sectional design and subjective data collection.
  4. Language:

    • Proofread for grammatical errors and awkward phrasing.
    • Use concise language and avoid repetition in the discussion and results.
    • Provide a glossary or explanation of technical terms to improve reader comprehension.
  5. Include more relevant articles - this is not a recommendation, or a must do it phase, but could help you deepen your knowledge in the area:
    - Busa, F., Csima, M. P., Márton, J. A., Rozmann, N., Pandur, A. A., Ferkai, L. A., Deutsch, K., Kovács, Á., & Sipos, D. (2023). Sleep Quality and Perceived Stress among Health Science Students during Online Education-A Single Institution Study. Healthcare (Basel, Switzerland), 12(1), 75. https://doi.org/10.3390/healthcare12010075

    - Sipos, D., Goyal, R., & Zapata, T. (2024). Addressing burnout in the healthcare workforce: current realities and mitigation strategies. The Lancet regional health. Europe, 42, 100961. https://doi.org/10.1016/j.lanepe.2024.100961

Author Response

Reviewer 2

Thank you for having me as a reviewer for this article - here are my suggestions:

Strengths:

  • The study employs a clear and appropriate cross-sectional design to explore the associations between health promotion behaviors, occupational burnout, and sleep disturbances.
  • The use of validated instruments, such as the Athens Insomnia Scale (CAIS-8) and the Chinese Occupational Burnout Scale, adds credibility.
  • Ethical approval and informed consent were adequately addressed.

Weaknesses:

  • Comments 1: Subjectivity in Data Collection: The reliance on self-reported questionnaires introduces biases such as social desirability or recall bias, potentially affecting the accuracy of the results.
  • Response 1: Thank you for pointing this out. We agree with this comment. Therefore, we have added limitations of the study. This addition can be found on page 10, lines 442-455.
  • Revised Text in the Manuscript:

[Limitations and Recommendations: This study has three limitations. First, sleep disturbances were measured subjectively using self-reported questionnaires, without employing objective wearable devices to monitor sleep. Second, data were collected from a single site within a traditional factory, which limits the generalizability of the findings. Third, as a cross-sectional study, it only allows for the exploration of associations and does not enable the investigation of causal relationships.

Future studies should integrate objective methods, such as wearable devices or poly-somnography, to monitor sleep patterns and disturbances. These tools can offer more precise and comprehensive data, minimizing potential biases linked to self-reported measures. Additionally, to improve the generalizability of findings, future research should broaden the data collection to include multiple sites across various traditional industries. This would facilitate comparisons across different work environments and populations. Moreover, longitudinal studies are recommended to establish causal relationships among health promotion behaviors, occupational burnout, and sleep disturbances. By tracking changes over time, these studies can provide deeper insights into the dynamic interactions between these variables.]

  • Comments 2: Exclusion Criteria: The exclusion criteria (e.g., pregnant participants) are not adequately justified, potentially overlooking a significant subset of the population that might experience different patterns of sleep disturbances.
  • Response 2: We appreciate this insightful comment. According to the literature, pregnant individuals are more prone to experiencing sleep disturbances, which could potentially affect the results of this study. Therefore, they were excluded from participation. However, it is worth noting that no pregnant individuals were present in the included study sites.
  • Comments 3: Sample Representation: The study only considers employees from a specific sector in Northern Taiwan, which may limit the generalizability of findings to other industries or regions.
  • Response 3: Thank you for pointing this out. We agree with this comment. Therefore, we have added limit of the study. This addition can be found on page 10, lines 442-455.
  • Comments 4: Measurement Validity: While validated scales were used, the absence of objective sleep measurements (e.g., actigraphy) reduces the reliability of sleep disturbance data.
  • Response 4: Thank you for pointing this out. We agree with this comment. Therefore, we have added limit of the study. This addition can be found on page 10, lines 442-455.
  • Comments 5 Cluster Sampling: Random selection from one factory could lead to homogeneity in the sample, reducing variability and potentially biasing results.
  • Response 5: Thank you for pointing this out. We agree with this comment. Therefore, we have added limit of the study. This addition can be found on page 10, lines 442-455.

Conclusion

Strengths:

  • The conclusions align with the study's objectives, emphasizing the role of work-related factors and burnout in sleep disturbances.
  • Practical recommendations for occupational health nurses and employers are well-articulated.

Weaknesses:

  • Comments 6: Overgeneralization: Conclusions suggest a causal relationship between variables, despite the cross-sectional design, which cannot establish causality.
  • Response 6: Agree. We have revised.to emphasize this point. Discuss the changes made, providing the necessary explanation. This addition can be found on page 10, lines 434-440.
  • Revised Text in the Manuscript:

[In our study, a total of 365 employee data were collected, with 63.3% of the study subjects working in shifts and 47.9% experiencing sleep disturbances. Overall findings indicate that shift work, health promotion behaviors such as "health responsibility," "life appreciation," "healthy diet," and “personal burnout” are factors associated with sleep disturbance. In non-shift work group, personal burnout and health responsibility are factors associated with sleep disturbance. In shift work group, work-related burnout, life appreciation behavior and stress management are factors associated with sleep disturbance.]

  • Comments 7: Limited Depth in Recommendations: While workplace interventions are suggested, specific implementation strategies or evidence from similar successful programs are not provided.
  • Response 7: Agree. We have revised to emphasize this point. Discuss the changes made, providing the necessary explanation. We have provided references in the text to support the success of similar programs as evidence. This addition can be found on page 9, lines 348-357.
  • Revised Text in the Manuscript:

[A study, conducted among workers in an automobile parts manufacturing company, utilized the Difference-in-Difference (DID) method to examine changes in sleep duration and the prevalence of poor sleep quality among shift and non-shift workers before and after adjustments to the work schedule. In 2010, the company transitioned from a weekly rotating continuous two-shift system to a weekly rotating discontinuous two-shift system, reducing working hours per shift while maintaining two groups (day and night shifts). Despite these adjustments, the weekly rotating shift schedule and the number of workdays remained unchanged. Statistically significant improvements were observed in the experimental group during night shifts: daily sleep duration increased by +0.5 hours, the wake-after-sleep-onset rate decreased by -13.9%, and self-reported poor sleep quality reduced by -34.9% Quitting overnight work led to improved sleep health for shift workers].

  • Comments 8: Inconsistent Emphasis: The conclusion downplays certain significant findings, such as the role of shift work and long working hours, which should be given more prominence considering their strong statistical associations.
  • Response 8: Thank you for pointing this out. We agree with this comment. Therefore, we have added conclusion of the study. This addition can be found on page 9, lines 348-357.
  • Revised Text in the Manuscript:

[In our study, a total of 365 employee data were collected, with 63.3% of the study subjects working in shifts and 47.9% experiencing sleep disturbances. Overall findings indicate that shift work, health promotion behaviors such as "health responsibility," "life appreciation," "healthy diet," and “personal burnout” are factors associated with sleep disturbance. In non-shift work group, personal burnout and health responsibility are factors associated with sleep disturbance. In shift work group, work-related burnout, life appreciation behavior and stress management are factors associated with sleep disturbance.]

Language

Strengths:

  • The paper is well-organized, with clear headings and a logical flow between sections.
  • Technical terms are appropriately used, and references are cited correctly.

Weaknesses:

  • Comments 10: Repetitive Language: Some sections (e.g., discussion and results) repeat the same points without adding new insights.
  • Response 10: Thank you for pointing this out. We agree with this comment. Therefore, we have explained what changes you have made. Mention exactly where in the revised manuscript this change can be found page 9, lines 348-357, 362-363, 388-391, and page 10, lines 406-432.
  • Revised Text in the Manuscript:
  • [A study, conducted among workers in an automobile parts manufacturing company, utilized the Difference-in-Difference (DID) method to examine changes in sleep duration and the prevalence of poor sleep quality among shift and non-shift workers before and after adjustments to the work schedule. In 2010, the company transitioned from a weekly rotating continuous two-shift system to a weekly rotating discontinuous two-shift system, reducing working hours per shift while maintaining two groups (day and night shifts). Despite these adjustments, the weekly rotating shift schedule and the number of workdays remained unchanged. Statistically significant improvements were observed in the experimental group during night shifts: daily sleep duration increased by +0.5 hours, the wake-after-sleep-onset rate decreased by -13.9%, and self-reported poor sleep quality reduced by -34.9% Quitting overnight work led to improved sleep health for shift workers]
  • [People who prioritize health and nutrition tend to sleep better than those who completely neglect it, consistent with the findings of this study]
  • Enhance the discussion on non-significant results and propose hypotheses for future research. Consider how local culture and regulatory environments may influence the study outcomes and their applicability. [In Table 3, both groups with and without sleep disturbances showed low exercise scores. Future studies could explore the potential of encouraging regular physical activity to examine its relationship with sleep disturbances]
  • [Addressing burnout requires a comprehensive approach that includes effective workplace resource management and opportunities for skill development. At the employer level, establishing conditions that foster optimal work performance is essential. Providing emotional support and resilience training equips employees with valuable resources to manage stress. To improve job performance, optimizing workloads, reducing excessive working hours, and promoting task sharing are critical strategies. The integration of automation and artificial intelligence (AI) technologies further enhances work processes, reduces employee burnout, and boosts efficiency. Moreover, AI robots can be utilized to handle repetitive or demanding tasks, alleviating the burden on individuals and mitigating personal burnout. On a personal level, self-care is indispensable. Prioritizing physical fitness, ensuring adequate rest, and practicing effective stress management are vital to maintaining overall well-being and preventing burnout. Through these combined efforts, both organizations and individuals can create a healthier and more productive work environment.
  • Our study found that shift workers' work-related burnout, life appreciation behavior, and stress management are linked to sleep disturbances, while for non-shift workers, personal burnout and health responsibility are associated with sleep disturbances. Possible explanations include: shift workers face irregular work hours, disrupting their circadian rhythm, in-creasing physical and mental stress, and affecting sleep quality. Work-related burnout may reflect resource depletion due to long or night shifts. Non-shift workers, with more stable work hours, face pressures from balancing work and family responsibilities, leading to personal burnout, which affects their health responsibility and sleep. Non-shift workers' burnout often stems from the dual pressures of work and family, such as caring for children and aging parents. [80] This can hinder their ability to manage health, affecting sleep. In contrast, shift workers' primary challenge is irregular work and lifestyle, with less family-related pressure.
  • These differences highlight the need for tailored health promotion strategies. For shift workers, interventions should focus on stress management and life appreciation; for non-shift workers, strategies should address personal burnout and enhance health responsibility. Future research should explore the role of family and social factors in sleep disturbances.]
  • Comments 11: Grammatical Errors: There are occasional grammatical mistakes, such as "increases risks of occupational injuries" (should be "increased risks").
  • Response 11: Thank you for pointing this out. We have revised as follows: page 2, line 84.
  • Revised Text in the Manuscript: [Higher burnout levels increased risks of sleep disturbances]
  • Comments 12: Awkward Phrasing: Sentences like “employees’ burnout and sleep disturbances are mutual risk factors” could be rephrased for better readability.
  • Response 12: Thank you for pointing this out. We agree with this comment. We deleted it.
  • Comments 13: Overuse of Jargon: Simplifying technical terms or providing definitions for less familiar terms (e.g., "circadian rhythm disorders") could improve accessibility.
  • Response 13: Thank you for pointing this out. We have revised as follows: page number 9, lines 342-343.
  • Revised Text in the Manuscript: [Shift work is essential in modern industries due to economic demands but often leads to natural sleep-wake cycle disturbances,]

Recommendations for Improvement 

  1. Methodology:
  • Comments 14: Include objective measures for sleep disturbances, such as actigraphy or polysomnography.
  • Response 14: Thank you for your valuable suggestion. In this study, we primarily relied on self-reported questionnaires to assess sleep disturbances, which is a common and effective measurement method. However, we acknowledge that objective measures, such as behavioral assessments or polysomnography, can provide additional data support. This limitation has been addressed under "Limitation 1" in the manuscript. Future research will consider incorporating these objective measures to enhance the reliability and accuracy of the findings. This addition can be found on page 10, lines 442-455.
  • Comments 15: Broaden inclusion criteria and ensure diversity in the sample population.
  • Response 15: We appreciate the reviewer’s suggestion regarding sample diversity. In this study, we selected workers from a specific industry to focus on sleep disturbances within this occupational group. However, we agree that future research should aim to broaden the sample diversity by including individuals of different genders, ages, regions, and job types to enhance the external validity of the study. This approach would allow us to capture a more accurate picture of sleep disturbances across various demographic groups and improve the generalizability of the findings. However, due to the rarity of researchers gaining access to factory environments for studies, the current study was limited to a single site (plant 1) as the sample source. Moving forward, we recommend encouraging occupational health nurses within factories to expand the recruitment of participants to include a wider range of workers. This could significantly contribute to the robustness and applicability of future research findings. There are limitations of our study, and we have provided an explanation in the limitations section.
  • Comments 16: Justify the sampling method more explicitly to address potential biases.
  • Response 16: Thank you for pointing this out. We appreciate this insightful comment.
  1. The number of participants in Plant 1 met the minimum required sample size.
  2. The researchers used a chi-square test to analyze the average age of the parent population, which was 44.5 years (t = -1.03, p = 0.3), indicating no significant difference between the average age of Plant 1 and the parent population.
  3. The unit supervisor of Plant 1 agreed to proceed with the recruitment and has signed the project consent form.
  1. Data Analysis:
  • Comments 17: Address potential multicollinearity and include a discussion of effect sizes.
  • Response 17: Thank you for pointing this out. We agree with your comment and have made the necessary revisions to address it. The statistics have been re-analyzed, and the results have been updated accordingly. To avoid interaction effects, stepwise regression was used in Model 4. Please see page 8, lines 298-308.
  • Revised Text in the Manuscript:
  • [For Model 4, Stepwise logistic regression analysis was conducted using work patterns, health promotion behaviors, and occupational burnout as independent variables and sleep disturbances as the dependent variable. The results are as follows: personal burnout, OR = 1.07 (95% CI 1.06,1.09), p < 0.001 and shift work (Yes/No), OR = 2.67 (95% CI 1.53,4.65), p = 0.001; and. The results are as follows: health responsibility behavior, OR = 0.50 (95% CI 0.33,0.77), p = 0.001; life appreciation behavior, OR = 0.47 (95% CI 0.29,0.76), p = 0.002. These results indicate a higher likelihood of sleep disturbances associated with higher scores in personal burnout, shift work, lower scores in health responsibility behavior and life appreciation behavior. The explanatory power (R²) of this model was 0.45. See Table 4 for detailed results. A total of 365 subjects were collected. In terms of the proportion of sleep disturbances among shift and non-shift workers, the effect size is approximately 0.91.]
  • Comments 18: Perform subgroup analyses for demographic variables and report findings comprehensively.
  • Response 18: We appreciate the reviewer’s suggestion regarding subgroup analysis. In the bivariate analysis, factors such as age, gender, marital status, smoking, alcohol consumption, and education level showed no significant differences. However, shift work exhibited a significant difference. Therefore, we used shift work as a variable in the stepwise regression analysis, which revealed distinct influencing factors This addition can be found on Page 7, lines 269-277 (Table 4) Revised data and analysis have been incorporated to ensure clarity and accuracy.
  • Revised Text in the Manuscript:
  • [We included the independent variables with significant differences in the bivariate analysis into the logistic regression. For Model 1, logistic regression analysis was conducted using work patterns as independent variables and sleep disturbances as the dependent variable. The results are as follows: shift work (yes/no), OR = 2.79 (95% CI 1.74,4.49), p < 0.001; weekly working hours (41~48 hours/≤40 hours), OR = 2.23 (95% CI 1.37,3.65), p = 0.001; and weekly working hours (˃48 hours/≤40 hours), OR = 3.89 (95% CI 1.19,12.99), p = 0.027. These results indicate that higher frequencies of shift work and working more than 40 hours per week are associated with a higher likelihood of sleep disturbance. The explanatory power (R²) of this model was 0.09. See Table 4 for detailed results.]
  • Comments 19: Use sensitivity analyses to confirm the robustness of findings.
  • Response 19: We conducted subgroup analysis (shift work vs. non-shift work) to examine the consistency of the results as a sensitivity analysis. The results were consistent with the overall findings. We confirm the robustness of findings. Please see page 8, lines 309-320.
  • For Model 5, 6, since shift work is related to sleep disturbances, shift and non-shift workers were separated, and the work patterns were classified accordingly. Work patterns, health promotion behaviors, and occupational burnout were set as independent variables, and sleep disturbances as the dependent variable. A stepwise logistic regression analysis was conducted (to avoid interaction effects) to perform sensitivity analyses to confirm the robustness of findings. Non-shift work, the results are as follows: personal burnout, OR = 1.09 (95% CI 1.06,1.12), p < 0.001 and health responsibility, OR = 0.35 (95% CI 0.21,0.57), p < 0.001, the explanatory power (R²) of this model was 0.44.
    • Shift work, the results are as follows: work-related burnout OR = 1.07 (95% CI 1.04,1.10), p < 0.001, life appreciation behavior, OR = 0.27 (95% CI 0.11,0.66), p= 0.004, and stress management OR = 0.32 (95% CI 0.11,0.93), p =0.035, the explanatory power (R²) of this model was 0.47. See Table 5 for detailed results.
  1. Conclusions:
  • Comments 20:
    1. Avoid causal language and clarify that findings indicate associations, not causality.
    2. Expand on practical recommendations with specific examples of workplace health programs.
    3. Highlight limitations more transparently, especially those related to the cross-sectional design and subjective data collection.
  • Response 20: Thank you for your suggestion. We will pay attention to word choices. We have revised to emphasize this point. Discuss the changes made, providing the necessary explanation. This addition can be found on page 10, lines 434-440.
    • Revised Text in the Manuscript: [In our study, a total of 365 employee data were collected, with 63.3% of the study subjects working in shifts and 47.9% experiencing sleep disturbances. Overall findings indicate that shift work, health promotion behaviors such as "health responsibility," "life appreciation," "healthy diet," and “personal burnout” are factors associated with sleep disturbance. In non-shift work group, personal burnout and health responsibility are factors associated with sleep disturbance. In shift work group, work-related burnout, life appreciation behavior and stress management are factors associated with sleep disturbance.]

Language:

  • Comments 21:
    1. Proofread for grammatical errors and awkward phrasing.
    2. Use concise language and avoid repetition in the discussion and results.
    3. Provide a glossary or explanation of technical terms to improve reader comprehension.

Response 21: Thank you for your suggestion. I have provided the English editing certificate for this article on the last page of this reply.

  • Comments 22: Include more relevant articles - this is not a recommendation, or a must do it phase, but could help you deepen your knowledge in the area:
    - Busa, F., Csima, M. P., Márton, J. A., Rozmann, N., Pandur, A. A., Ferkai, L. A., Deutsch, K., Kovács, Á., & Sipos, D. (2023). Sleep Quality and Perceived Stress among Health Science Students during Online Education-A Single Institution Study. Healthcare (Basel, Switzerland), 12(1), 75. https://doi.org/10.3390/healthcare12010075

    - Sipos, D., Goyal, R., & Zapata, T. (2024). Addressing burnout in the healthcare workforce: current realities and mitigation strategies. The Lancet regional health. Europe, 42, 100961. https://doi.org/10.1016/j.lanepe.2024.100961
  • Response 22: First, we sincerely thank the reviewer for providing two articles that have helped us refine the structure and expression of our manuscript. The first article has been applied to page 9, lines 362-363 and page 9, lines 388-391. The second article has been incorporated into page 10, lines 406-417.

Reviewer 3 Report

Comments and Suggestions for Authors

Thank you very much for the opportunity to review the article "Exploring health-promoting behaviors, occupational burnout, and sleep disorders in traditional industry workers”.

This is a very important article for the scientific community and, particularly, for occupational health teams. The article is robust, very well organized, and of excellent scientific quality, but it is necessary to clarify/correct a few small things:

- The methodology does not present the objectives of the study - it should be

- Section 2.3, in the Inclusion Criteria, there is no criterion “being a worker”; was this variable controlled? The two criteria mentioned do not respond to the characteristics of the study

- Section 2.4 states that 375 questionnaires were distributed and 365 were returned. It should be clear how the questionnaires were distributed, how people were prepared to fill out the questionnaires. How were they returned and delivered to whom? Why were the 10 questionnaires not returned? What happened?

Best regards

Author Response

Reviewer3

Thank you for your recognition and encouragement.

Comment 1: The methodology does not present the objectives of the study - it should be.

Response 1: Thank you for pointing this out. We agree with this comment. Therefore, we have added objectives of the study.

This addition can be found on page 2, lines 106-108.

Revised Text in the Manuscript:

“The objective of this study [Therefore, this study aims to investigate the correlation among work patterns, health promotion behaviors, occupational burnout, and sleep disturbances among employees in traditional industries.].

Comment 2: Section 2.3, in the Inclusion Criteria, there is no criterion “being a worker”; was this variable controlled? The two criteria mentioned do not respond to the characteristics of the study.

Response 2: Thank you for your valuable observation. We have clarified the inclusion criteria to address this issue. The criterion " 1. Currently employed staff with at least three months of experience in traditional industries.

This revision is located on page 3, lines 122-123.

Revised Text in the Manuscript:
“The inclusion criteria for this study are: [1. Currently employed staff with at least three months of experience in in traditional industries]

Comment 3: Section 2.4 states that 375 questionnaires were distributed and 365 were returned. It should be clear how the questionnaires were distributed, how people were prepared to fill out the questionnaires. How were they returned and delivered to whom? Why were the 10 questionnaires not returned? What happened?

Response 3: We appreciate this insightful comment.

During the study period of one month, 375 questionnaires were distributed, enclosed in sealed envelopes along with an information sheet, and handed out by the factory nurse to employees who had been working for more than three months. Participants were instructed to read the materials independently and could request guidance via telephone if needed. Completed questionnaires were returned to the factory nurse’s office, where the researcher checked them for missing information or sections requiring clarification. Upon completion, participants were given small gifts as a token of appreciation. Ten employees did not return the questionnaires. Since the process was anonymous, it was not possible to follow up on the reasons for non-return.

Reviewer 4 Report

Comments and Suggestions for Authors

1. Lines 24-28 The numerical results of the study should be presented.

2. Table 1. Descriptive statistics of basic attributes. The characteristics (Gender, Marital Status, Smoking Habits, Number of Cigarettes, Drinking Habits, etc.) should be explained to make understandable how relevant these data are for this study

3.  Lines 437-443 Author Contributions should be presented according to the journal template.

Author Response

Reviewer 4

Comment 1: Lines 24-28 The numerical results of the study should be presented.

Response 1: Thank you for your valuable observation. This revision is located on page 1, lines 21-24. 

Revised Text in the Manuscript: 

“[Factors associated with sleep disturbances included personal burnout, OR = 1.07 (95% CI 1.06,1.09), p < 0.001 and shift work, OR = 2.67 (95% CI 1.53,4.65), p < 0.001; and health responsibility behavior, OR = 0.50 (95% CI 0.33,0.77), p = 0.001; life appreciation behavior, OR = 0.47 (95% CI 0.29,0.76), p = 0.002.] " 

Comment 2: Table 1. Descriptive statistics of basic attributes. The characteristics (Gender, Marital Status, Smoking Habits, Number of Cigarettes, Drinking Habits, etc.) should be explained to make understandable how relevant these data are for this study.

Response 2: Thank you for your observation. The correlation between basic demographic data and sleep debt has been included in Table 3.

This addition can be found on page 6, lines 259-264.

Revised Text in the Manuscript:

“[Among the participants, 48.7% of males and 35% of females experienced sleep disturbances. Sleep disturbances were reported by 51.4% of those with a partner and 45.7% of those without, 51.0% of smokers and 46.8% of non-smokers, 48.9% of drinkers and 47.6% of non-drinkers, as well as 48.8% of those with a high school education or lower and 47.1% of those with a college education or higher. However, none of these variables demonstrated a statistically significant correlation.].”

Comment 3: Lines 437-443 Author Contributions should be presented according to the journal template.

Response 3: Thank you for pointing this out. We appreciate your insightful comment and have made the following revision to address it.

This addition is now included on page 11, lines 457-461, as follows:

Revised Text in the Manuscript:

"The characteristics [Conceptualization, Y.F.Y., Y.Y.C., and S.H.C.; methodology, Y.F.Y. and S.H.C.; formal analysis, Y.F.Y. and S.H.C.; investigation, Y.F.Y.; writing—original draft preparation, Y.F.Y., Y.Y.C., and S.H.C.; writing—review and editing, Y.F.Y., Y.Y.C., and S.H.C.; visualization, Y.F.Y. and S.H.C.; supervision, S.H.C.; project administration, Y.F.Y. and S.H.C. All authors have read and agreed to the published version of the manuscript]."

Reviewer 5 Report

Comments and Suggestions for Authors

The Abstract is not informative. Make it precise and coherent.

The response rate is 100%. How practical was this in your setting? What makes all the participant agreed for the interview?

Introduction:

It lacks coherence.

The gap is mentioned in the middle (line 65-71). It should align with the lines 97-106.

Method:

-          2.2, study setting: do not tell significance of the study (line 117-118)

-          2.3, why are pregnant excluded from the study?

-          2.4, Why cluster sampling is employed if all the factories share similar characteristics such as background and work nature?

-          How do you think this study is free of bias while you provide lunch bag or multifunctional massager? Is this the reason for 100% response rate?

-          2.5, Measures: You use likert scale for the response variables. How do you precisely assure the response is in a particular scale? How do you control a response bias? The workers may provide you desirable response expecting some interventions or fear of their job security?

Result

-          Table 2, How do you relate the higher personal burn out with some social aspects in his/her family?

Limitation: Limitation can be discussed in detail. However, the presentations mix limitation and recommendation. This needs improvement.

Comments on the Quality of English Language

English language needs improvement. 

Author Response

Reviewer 5

Comment 1: The Abstract is not informative. Make it precise and coherent.

Response 1: Thank you for pointing this out. We agree with this comment. We have revised the abstract content, please see abstract.

Revised Text in the Manuscript:

“The Abstract of this study [Sleep disturbances affect about 40% of the global population and are a common issue among patients seeking medical consultation. There is limited research on sleep disturbances in Taiwan's traditional industry workforce. This study aims to investigate the correlation among work patterns, health promotion behaviors, occupational burnout, and sleep disturbances among employees in traditional industries. A cross-sectional study was conducted to collect data on the work patterns, health promotion behavior, occupational burnout, and sleep disturbances from a traditional industry. The study period was from May to June 2023. Data analysis was performed using chi-square tests, independent sample t-tests, and logistic regression. A total of 365 employee data were collected, with 63.3% of the study subjects working in shifts and 47.9% experiencing sleep disturbances. Factors associated with sleep disturbances included personal burnout, OR = 1.07 (95% CI 1.06,1.09), p < 0.001 and shift work, OR = 2.67 (95% CI 1.53,4.65), p < 0.001; and health responsibility behavior, OR = 0.50 (95% CI 0.33,0.77), p = 0.001; life appreciation behavior, OR = 0.47 (95% CI 0.29,0.76), p = 0.002. Occupational health nurses should regularly assess employees' sleep status and provide psychological counseling services and health promotion pro-grams to help employees alleviate sleep disturbances.”

Comment 2: The response rate is 100%. How practical was this in your setting? What makes all the participants agree for the interview?

Response 2: Thank you for pointing this out.

We distributed 375 questionnaires and received 365 responses, achieving a high response rate of 97.3%. The primary reason for this high response rate is that many employees were experiencing sleep-related issues, and both supervisors and the factory nurse actively encouraged employees to complete the questionnaire.

Comment 3: Introduction:

*It lacks coherence.

*The gap is mentioned in the middle (line 65-71). It should align with the lines 97-106.

Response 3: Thank you for your valuable observation. We have accordingly revised to this point. Mention exactly where in the revised manuscript this change can be found - page 2, lines 61-73.

Revised Text in the Manuscript:

”[Studies have shown that sleep disturbances can lead to increased stress, job insecurity, and poor job performance, contributing to absenteeism [18,20–21]. Recent studies have identified several factors influencing sleep disturbances among employees, including background characteristics such as being female, older age [23,24], smoking, lower educational attainment, alcohol use, and chronic illnesses [23-26]. While these are established risk factors, prior research has primarily focused on non-traditional industry workers. Given that work patterns in traditional industries differ significantly, the factors affecting sleep disturbances in these settings may also vary. Manufacturing remains a major industry in many countries, but previous research on manufacturing employees has largely centered on the health risks associated with occupational exposure [27]. Early identification of the prevalence and contributing factors of sleep disturbances is crucial, as it not only improves employee health and quality of life but also helps reduce national healthcare costs. Despite its importance, there is a notable lack of studies examining sleep disturbances and their influencing factors among traditional manufacturing industry employees in Taiwan.]”

Comment 4: Method:

*2.2, study setting: do not tell significance of the study (line 117-118)

Response: Thank you for your valuable observation. We have accordingly changed to this point. Mention exactly where in the revised manuscript this change can be found  page 3, lines 117-119.

Revised Text in the Manuscript:

[Study setting was at a traditional industrial factory in northern Taiwan primarily engaged in the textile industry, rubber and plastic raw material manufacturing, and plastic products manufacturing.]

*2.3, why are pregnant excluded from the study?

Response: We appreciate this insightful comment.

According to the literature, pregnant individuals are more prone to experiencing sleep disturbances, which could potentially affect the results of this study. Therefore, they were excluded from participation. However, it is worth noting that no pregnant individuals were present in the included study sites.

*2.4, Why cluster sampling is employed if all the factories share similar characteristics such as background and work nature?

Response: Thank you for your valuable observation. We have accordingly revised to this point. Mention exactly where in the revised manuscript this change can be found - page 3, lines 126-129.

Revised Text in the Manuscript:

The study participants were employees from a large plastic product manufacturing company located in northern Taiwan. This traditional industrial factory operates five plants, those plants all involve labor-intensive work. Due to the hazardous nature of the plants, the factory only allowed us to conduct the research in plant 1.

*.How do you think this study is free of bias while you provide lunch bag or multifunctional massager? Is this the reason for 100% response rate?

Response: Thank you for pointing this out. Mention can be found page 3, lines 133-134.

We distributed a total of 375 questionnaires and collected 365 completed questionnaires, achieving a response rate of 97.3%. Lunch bags or multifunctional massagers were only distributed after the questionnaires were collected. Therefore, the distribution of gifts should not influence the results.

*2.5, Measures: You use likert scale for the response variables. How do you precisely assure the response is in a particular scale? How do you control a response bias? The workers may provide you desirable response expecting some interventions or fear of their job security?

Response: Thank you for your valuable observation. We have accordingly revised to this point. Mention exactly where in the revised manuscript this change can be found page 4, lines 184-185.

Revised Text in the Manuscript:

[The Cronbach's alpha for this scale was 0.82 to 0.84 and our study was 0.89. In our study, after two weeks, the test-retest reliability coefficient for self-reported sleep disturbances was 0.82.]

Comment 5: Result

*. Table 2, How do you relate the higher personal burn out with some social aspects in his/her family?

Response 5: Thank you for pointing this out. Mention can be found page 10, lines 418-432.

Our study found that shift workers' work-related burnout, life appreciation behavior, and stress management are linked to sleep disturbances, while for non-shift workers, personal burnout and health responsibility are associated with sleep disturbances. Possible explanations include: shift workers face irregular work hours, disrupting their circadian rhythm, increasing physical and mental stress, and affecting sleep quality. Work-related burnout may reflect resource depletion due to long or night shifts. Non-shift workers, with more stable work hours, face pressures from balancing work and family responsibilities, leading to personal burnout, which affects their health responsibility and sleep. Non-shift workers' burnout often stems from the dual pressures of work and family, such as caring for children and aging parents. This can hinder their ability to manage health, affecting sleep. In contrast, shift workers' primary challenge is irregular work and lifestyle, with less family-related pressure.

These differences highlight the need for tailored health promotion strategies. For shift workers, interventions should focus on stress management and life appreciation; for non-shift workers, strategies should address personal burnout and enhance health responsibility. Future research should explore the role of family and social factors in sleep disturbances.

*.Limitation: Limitation can be discussed in detail. However, the presentations mix limitation and recommendation. This needs improvement.

Response: Thank you for your valuable observation. We have accordingly revised to this point. Mention exactly where in the revised manuscript this change can be found page 10, lines 442-455.

Revised Text in the Manuscript:

Limitations and Recommendations

This study has three limitations. First, sleep disturbances were measured subjectively using self-reported questionnaires, without employing objective wearable devices to monitor sleep. Second, data were collected from a single site within a traditional factory, which limits the generalizability of the findings. Third, as a cross-sectional study, it only allows for the exploration of associations and does not enable the investigation of causal relationships.

Future studies should integrate objective methods, such as wearable devices or poly-somnography, to monitor sleep patterns and disturbances. These tools can offer more precise and comprehensive data, minimizing potential biases linked to self-reported measures. Additionally, to improve the generalizability of findings, future research should broaden the data collection to include multiple sites across various traditional industries. This would facilitate comparisons across different work environments and populations. Moreover, longitudinal studies are recommended to establish causal relationships among health promotion behaviors, occupational burnout, and sleep disturbances. By tracking changes over time, these studies can provide deeper insights into the dynamic interactions between these variables.

Round 2

Reviewer 1 Report

Comments and Suggestions for Authors

Thank you for your responses to the previous review report. Your clarifications and revisions have significantly improved the manuscript, making it much clearer and more comprehensible. Below, I provide additional feedback and suggestions to further enhance the quality and presentation of your study.

1. Study Settings and Participants:

The description of the study settings and participants is confusing and inconsistent. Was the study conducted in a single factory operating in four sectors or five different types of production? The section on study settings and participants needs to be rewritten for better clarity and correlation.

Example:

“Study setting was at a traditional industrial factory in northern Taiwan primarily engaged in the textile industry, rubber and plastic raw material manufacturing, and plastic products manufacturing.”

“The study participants were employees from a large plastic product manufacturing company located in northern Taiwan. This traditional industrial factory operates five plants, all involving labor-intensive work. Due to the hazardous nature of the plants, the factory only allowed the research to be conducted in Plant 1.”

Recommendation: Clarify whether the study covers multiple sectors or focuses on a specific one, and ensure consistency between the setting and participant descriptions.

2. Demographic Attributes:

It would be beneficial to rephrase and clarify the demographic and work-related variables included in your questionnaire as “based on our review of the literature, the demographic section of the questionnaire encompasses variables such as gender, age, education level, marital status, smoking habits, alcohol consumption, shift work, and weekly working hours.”

3. You mention using four models but explain six. This inconsistency needs to be addressed. Why were these additional models (5 and 6) included, and what was their purpose?

 “The analysis was divided into four primary models:

Model 1: Relationship between work patterns and sleep disturbances.

Model 2: Relationship between health promotion behaviors and sleep disturbances.

Model 3: Relationship between occupational burnout and sleep disturbances.

Model 4: Combined relationships between work patterns, health promotion behaviors, occupational burnout, and sleep disturbances.”

Models 5 and 6, which examine the relationship between weekly working hours, health promotion behaviors, occupational burnout, and sleep disturbances (with and without shift work), need additional justification.

4.    Include all explanatory powers as a row in Table 4 for better clarity and comparison.

5.    Just remove “See Table 4 for detailed results.” from every Model’s explanation,in the text, just change it as “(Model4)”

6.    For Table 5, clearly show all variables included in the models for both shift and non-shift workers. Indicate variables as "Not Applicable" where the stepwise model does not include them. This will improve clarity and comprehensiveness.

7.    Discussion – Summarize the first paragraph of the discussion. Start with your key findings and their significance compared to other studies.

8.    Example (it is not a good on efor your study just as an example) :

“Our study highlights that sleep disturbances are prevalent among study group working in traditional sectors, comparable to those reported in white-collar occupations. We found that, work-related burnout, life appreciation behaviors, and stress management were significant predictors of sleep disturbances for shift workers, , whereas for non-shift workers, personal burnout and health responsibility played a crucial role.”

9.    Reduce the number of numerical comparisons just summarise the points you conmpare made between your findings and existing literature.

10. Avoid summarizing other studies in the discussion. Instead, integrate their findings with your own or make recommendations based on them.For example, the following paragraph could be rephrased:

“A study conducted among workers in an automobile parts manufacturing company used the Difference-in-Differences (DID) method to examine changes in sleep duration and quality after adjustments to work schedules. This study aligns with our findings, suggesting that reducing working hours per shift and minimizing night shifts can significantly improve sleep health.”

11. Health Promotion and Sleep Disturbances Section: Summarize the section on health promotion behaviors and sleep disturbances. Focus on how these behaviors relate to your findings and provide actionable insights. Example:

“Our study underscores the importance of health promotion behaviors such as life appreciation, healthy eating, and health responsibility in reducing sleep disturbances. Interventions promoting a positive mindset, improved nutrition (e.g., a Mediterranean diet), and proactive health management could benefit employees’ sleep quality and overall well-being.” With references of course

12. Do not include numerical data in the conclusion. Instead, provide a concise summary of your findings and their implications.

It would be more understandable if you have show all the variables you had put into the model for both shift and non-shift workers in Table 5 and mention as not applicale fort he variables if the stepwise model not calculated it

1.    Summarise the first paragraph of the discussion, start with your striking findings (not exactly the findings just cexplain how it is important when compared with other studies” . Ex. “Our study found that sleep disturbances are high as whitecollars in blue collars and shift workers' work-related burnout, life appreciation behavior, and stress management are linked to sleep disturbances, while for non-shift workers, personal burnout and health responsibility are associated with sleep disturbances.

2.    Decrease the numbers you have used for comparisons of your findings with the literatüre

3.    This is a summary of another study in the beginning of the discussion which is not proper, just relate it with your study, compare or make a recommendation based on it “A study, conducted among workers in an automobile parts manufacturing company, utilized the Difference-in-Dif- 369
ference (DID) method to examine changes in sleep duration and the revalence of poor sleep quality among shift and non-shift workers before and after adjustments to the work schedule. In 2010, the company transitioned from a weekly rotating continuous two-shift system to a weekly rotating discontinuous two-shift system, reducing working hours per
shift while maintaining two groups (day and night shifts). Despite these adjustments, the weekly rotating shift schedule and the number of workdays remained unchanged. Statistically significant improvements were observed in the experimental group during night shifts: daily sleep duration increased by +0.5 hours, the wake-after-sleep-onset rate decreased by -13.9%, and self-reported poor sleep quality reduced by 34.9% Quitting over-night work led to improved sleep health for shift workers [59].

4.    Summarise this section, just discuss with your findings ““Sleep is one of the three pillars of health, alongside nutrition and exercise, and is seen 380
as a modifiable goal for well-being. To address rising healthcare costs, innovative employ ers are investing in their workforce through Worksite Health Promotion programs [60]. Model 2 found that behaviors related to "health responsibility," "appreciation of life," and "healthy eating" were linked to sleep disturbances. People who prioritize health and nu- trition tend to sleep better than those who completely neglect it, consistent with the find- ings of this study [61] According to Merrill et al. (2011) [62], participation in health pro- grams leads to improvements in exercise, diet, and stable sleep patterns, enhancing health and life satisfaction. Such programs help employees develop healthy behaviors, emphasizing the role of workplace nurses in promoting well-being. Weitzer et al. (2021) [63] observed that practicing "appreciation of life" behaviors
such as maintaining an optimistic outlook, can improve sleep quality and reduce disturbances, consistent with this study's findings. The questionnaire on "Life appreciation behaviors" included items like "maintaining a smile or laughter every day" and "appreciating people, events, and nature." Thus, fostering a positive mindset, frequent smiles, 394
self-appreciation, and spending time outdoors are recommended to lower sleep disturbance risks. This study supports findings by Cao et al. (2020) [64] and Castro-Diehl et al. (2018) [65] that "healthy eating behavior" is linked to sleep disturbances. Castro-Diehl et al. (2018) [65] also found that a healthy diet can promote better sleep duration and reduce insomnia 399
risk. It is recommended to consult nutritionists to adjust employee cafeteria meals toward a Mediterranean diet (featuring healthy fats, moderate fish and poultry, limited red meat) and whole grains [64,66], promoting healthy eating habits among employees. "Health responsibility" involves a proactive attitude towards one's well-being, including seeking health information and monitoring physical and mental states. Higher "health responsibility" scores were associated with fewer sleep disturbances, likely due to better self-management. Future studies could explore this relationship further.
Although previous research Xie et al. (2021) [67] and Bjornsdottir et al. (2024) [68] indicates that regular exercise improves sleep quality and reduces insomnia, this study did not find a clear link, possibly because most participants did not engage in regular exercise. Encouraging regular physical activity, especially for shift workers, could improve sleep quality [69,70]. Future initiatives might include smart bands, stair-climbing campaigns, and accessible workplace exercise options, with follow-up studies to confirm exercise's benefits for sleep disturbances. In Table 3, both groups with and without sleep disturbances showed low exercise scores. Future studies could explore the potential of encouraging regular physical activity to examine its relationship with sleep disturbances [61]

5.       Don’t use numbers in conclusion

Author Response

Dear reviewer,

Thank you so much for the suggestions and we followed your comments and revised the paper again. Please see the responses as below.

Comments 1: 1. Study Settings and Participants:

The description of the study settings and participants is confusing and inconsistent. Was the study conducted in a single factory operating in four sectors or five different types of production? The section on study settings and participants needs to be rewritten for better clarity and correlation.

Example:

“Study setting was at a traditional industrial factory in northern Taiwan primarily engaged in the textile industry, rubber and plastic raw material manufacturing, and plastic products manufacturing.”

“The study participants were employees from a large plastic product manufacturing company located in northern Taiwan. This traditional industrial factory operates five plants, all involving labor-intensive work. Due to the hazardous nature of the plants, the factory only allowed the research to be conducted in Plant 1.”

Recommendation: Clarify whether the study covers multiple sectors or focuses on a specific one, and ensure consistency between the setting and participant descriptions.

Response 1: Thank you for pointing this out. We agree with this comment. We have revised text in the manuscript, please see page 3, lines 127-129, 136-138.

[Study setting was at a traditional industrial factory in northern Taiwan primarily engaged in plastic raw material and plastic products manufacturing. There are five plants in the traditional industrial factory.]

[The study participants were selected using purposive sampling and conducted in the traditional industrial factory. Following discussions with department management regarding safety considerations, the research was confined to plant 1.]

Comments 2. Demographic Attributes:

It would be beneficial to rephrase and clarify the demographic and work-related variables included in your questionnaire as “based on our review of the literature, the demographic section of the questionnaire encompasses variables such as gender, age, education level, marital status, smoking habits, alcohol consumption, shift work, and weekly working hours.”

Response 2: Thank you for your valuable observation. We have accordingly revised to this point. Mention exactly where in the revised manuscript this change can be found - page 4, lines 151-154.

[Demographic attributes included basic attributes and work patterns. Based on literatures, the basic attributes of the questionnaire encompassed variables such as gender, age, education level, marital status, smoking habits, and alcohol consumption. Work patterns included shift work, weekly working hours, and managerial position [23–26, 28–30].]

Comments 3: You mention using four models but explain six. This inconsistency needs to be addressed. Why were these additional models (5 and 6) included, and what was their purpose?

 “The analysis was divided into four primary models:

Model 1: Relationship between work patterns and sleep disturbances.

Model 2: Relationship between health promotion behaviors and sleep disturbances.

Model 3: Relationship between occupational burnout and sleep disturbances.

Model 4: Combined relationships between work patterns, health promotion behaviors, occupational burnout, and sleep disturbances.”

Models 5 and 6, which examine the relationship between weekly working hours, health promotion behaviors, occupational burnout, and sleep disturbances (with and without shift work), need additional justification.

Response 3: Thank you for pointing this out. We agree with your comment. Shifts are an unavoidable factor in our research, so we used Models 5 and 6 to explore their related factors. Our study found that work-related burnout, life appreciation behavior, and stress management are significant predictors of sleep disturbances for shift workers, while personal burnout and health responsibility play key roles for non-shift workers. Shift workers face irregular hours that disrupt circadian rhythms, increase stress, and impair sleep quality, with burnout reflecting resource depletion from long or night shifts. Non-shift workers, with stable hours, often experience burnout from balancing work and family responsibilities, affecting their health management and sleep. While shift workers' challenges stem from irregular schedules, non-shift workers face greater family-related pressures. These differences highlight the need for tailored health promotion strategies. For shift workers, interventions should focus on stress management and life appreciation; for non-shift workers, strategies should address personal burnout and enhance health responsibility. Future research should explore the role of family and social factors in sleep disturbances.

Comments 4: Include all explanatory powers as a row in Table 4 for better clarity and comparison.

Response 4: Thank you for your valuable observation. We have accordingly revised to this point. Mention exactly where in the revised manuscript this change can be found - page 8-9 table 4.

Comments 5: Just remove “See Table 4 for detailed results.” from every Model’s explanation, in the text, just change it as “(Model4)”

Response 5: Thank you for your valuable observation. We have accordingly revised to this point. Mention exactly where in the revised manuscript this change can be found - lines 294, 304, 310, 321, 337, and 344.

Comments 6: For Table 5, clearly show all variables included in the models for both shift and non-shift workers. Indicate variables as "Not Applicable" where the stepwise model does not include them. This will improve clarity and comprehensiveness.

Response 6: Thank you for your valuable observation. We have accordingly revised to this point. Mention exactly where in the revised manuscript this change can be found - page 9-10, lines 322~344.

Revised Text in the [Manuscript: For Model 5, 6 since shift work is related to sleep disturbances, non-shift workers (Model 5, n=231) and shift (Model 6, n=134) were separated. and the work patterns were classified accordingly. Work patterns, health promotion behaviors, and occupational burnout were set as independent variables, and sleep disturbances as the de-pendent variable. A stepwise logistic regression analysis was conducted (to avoid interaction effects) to perform sensitivity analyses to confirm the robustness of findings. All variables included in the models for both shift and non-shift workers were weekly working, health promotion behaviors (stress management behaviors, physical activity behaviors, health responsibility behaviors, life appreciation behaviors, healthy eating behaviors, oral hygiene behaviors and occupational burnout (personal burnout, work-related burnout, overcommitted to work).Non-shift work, the results are as follows: personal burnout, OR = 1.09 (95% CI 1.06,1.12), p < 0.001 and health responsibility, OR = 0.35 (95% CI 0.21,0.57), p < 0.001, the explanatory power (R²) of this model was 0.44. Not Applicable were weekly working, health promotion behaviors (stress management behaviors, physical activity behaviors, life appreciation behaviors, healthy eating behaviors, oral hygiene behaviors and occupational burnout (work-related burnout, overcommitted to work). (Model 5)

Shift work, the results are as follows: work-related burnout OR = 1.07 (95% CI 1.04,1.10), p < 0.001, life appreciation behavior, OR = 0.27 (95% CI 0.11,0.66), p= 0.004, and stress management OR = 0.32 (95% CI 0.11,0.93), p =0.035, the explanatory power (R²) of this model was 0.47. Not Applicable were weekly working, health promotion behaviors (physical activity behaviors, health responsibility behaviors, healthy eating behaviors, oral hygiene behaviors and occupational burnout (personal burnout, over-committed to work). (Model 6)

Comments 7: Discussion – Summarize the first paragraph of the discussion. Start with your key findings and their significance compared to other studies.

Response 7: Thank you for pointing this out. We agree with this comment. We have revised text in the manuscript, please see page 10, lines 346-350.

Comments 8: Example (it is not a good on efor your study just as an example)

“Our study highlights that sleep disturbances are prevalent among study group working in traditional sectors, comparable to those reported in white-collar occupations. We found that, work-related burnout, life appreciation behaviors, and stress management were significant predictors of sleep disturbances for shift workers, where as for non-shift workers, personal burnout and health responsibility played a crucial role.”

Response 8: Thank you for pointing this out. We agree with this comment. We have revised text in the manuscript, please see page number 10, lines 346~350.

Comments 9: Reduce the number of numerical comparisons just summarise the points you compare made between your findings and existing literature. 9.

Response 9: Thank you for pointing this out. We agree with this comment. We have revised text in the manuscript, please see page number 10, lines 351~397.

Comments 10: Avoid summarizing other studies in the discussion. Instead, integrate their findings with your own or make recommendations based on them. For example, the following paragraph could be rephrased: 10.

“A study conducted among workers in an automobile parts manufacturing company used the Difference-in-Differences (DID) method to examine changes in sleep duration and quality after adjustments to work schedules. This study aligns with our findings, suggesting that reducing working hours per shift and minimizing night shifts can significantly improve sleep health.”

Response 10: Thank you for pointing this out. We agree with this comment. We have revised text in the manuscript, please see page 10-11, lines 377~384.

Revised Text in the Manuscript: Our findings are consistent with prior research conducted among workers in an automobile parts manufacturing company [59], which demonstrated that reducing working hours per shift and minimizing night shifts can significantly improve sleep health. These results underscore the importance of implementing similar interventions in traditional industrial settings, such as optimizing shift schedules to reduce physical and mental strain on employees. For instance, policies aimed at limiting prolonged night shifts or promoting shorter shifts could be effective strategies to mitigate sleep disturbances and enhance overall well-being.

Comments 11:  Health Promotion and Sleep Disturbances Section: Summarize the section on health promotion behaviors and sleep disturbances. Focus on how these behaviors relate to your findings and provide actionable insights.

“Our study underscores the importance of health promotion behaviors such as life appreciation, healthy eating, and health responsibility in reducing sleep disturbances. Interventions promoting a positive mindset, improved nutrition (e.g., a Mediterranean diet), and proactive health management could benefit employees’ sleep quality and overall well-being.” With references of course“

Response 11: Thank you for pointing this out. We agree with this comment. We have revised text in the manuscript, please see page number 11, lines 385~397.

Revised Text in the Manuscript: Sleep is one of the three pillars of health, alongside nutrition and exercise, and is seen as a modifiable goal for well-being. To address rising healthcare costs, innovative employers are investing in their workforce through Worksite Health Promotion pro-grams [60]. Our study highlights the critical role of health promotion behaviors, including life appreciation, healthy eating, and health responsibility, in mitigating sleep disturbances. Employees who maintain a positive mindset, such as practicing daily gratitude and optimism, report better sleep quality [63]. Additionally, adopting healthier eating habits, such as a Mediterranean diet rich in whole grains, healthy fats, and lean proteins, can further improve sleep patterns and reduce sleep disturbances risk [61,64~66]. Proactive health management, reflected in higher health responsibility scores, also enhances self-regulation and overall well-being. These findings emphasize the need for workplace interventions, such as nutrition counseling and wellness pro-grams, to foster these behaviors and promote better sleep health among employees [62]

Comments 12: Do not include numerical data in the conclusion. Instead, provide a concise summary of your findings and their implications.

Response 12: Thank you for pointing this out. We agree with this comment. We have revised text in the manuscript, please see page number 12, lines 455~ 464.

Revised Text in the Manuscript: [This study explored the factors associated with sleep disturbances among employees in traditional industries, with a particular focus on differences between shift and non-shift workers. The findings indicate that sleep disturbances are influenced by shift work and specific health promotion behaviors, including health responsibility, life appreciation, and healthy eating, as well as personal and work-related burnout. For non-shift workers, personal burnout and health responsibility are key factors affecting sleep quality. In contrast, for shift workers, work-related burnout, life appreciation behaviors, and stress management have a more significant impact on sleep quality. These findings highlight the unique challenges faced by each group and underscore the importance of tailored interventions to address their specific needs.]

Comments 13: It would be more understandable if you have shown all the variables you had put into the model for both shift and non-shift workers in Table 5 and mention as not applicable fort the variables if the stepwise model not calculated it1.

Response 13: Thank you for pointing this out. We agree with this comment. We have revised text in the manuscript, please see page 8-9, table 4.

Comments 14: Summarise the first paragraph of the discussion, start with your striking findings (not exactly the findings just explain how it is important when compared with other studies” . Ex. “Our study found that sleep disturbances are high as white collars in blue collars and shift workers' work-related burnout, life appreciation behavior, and stress management are linked to sleep disturbances, while for non-shift workers, personal burnout and health responsibility are associated with sleep disturbances.

Response 14: Thank you for pointing this out. We agree with this comment. We have revised text in the manuscript, please see page number 10, lines 346~ 350.

Comments 15: Decrease the numbers you have used for comparisons of your findings with the literature

Response 15: Thank you for pointing this out. We agree with this comment. We have revised text in the manuscript, please see page number 10~11, lines 351~ 397.

Comments 16: This is a summary of another study in the beginning of the discussion which is not proper, just relate it with your study, compare or make a recommendation based on it “A study, conducted among workers in an automobile parts manufacturing company, utilized the Difference-in-Dif- 369 ference (DID) method to examine changes in sleep duration and the revalence of poor sleep quality among shift and non-shift workers before and after adjustments to the work schedule. In 2010, the company transitioned from a weekly rotating continuous two-shift system to a weekly rotating discontinuous two-shift system, reducing working hours per shift while maintaining two groups (day and night shifts). Despite these adjustments, the weekly rotating shift schedule and the number of workdays remained unchanged. Statistically significant improvements were observed in the experimental group during night shifts: daily sleep duration increased by +0.5 hours, the wake-after-sleep-onset rate decreased by -13.9%, and self-reported poor sleep quality reduced by 34.9% Quitting over-night work led to improved sleep health for shift workers [59].

Response 16: Thank you for pointing this out. We agree with this comment. We have revised text in the manuscript, please see page 10-11, lines 377~384.

Comments 17:  Summarise this section, just discuss with your findings “Sleep is one of the three pillars of health, alongside nutrition and exercise, and is seen 380
as a modifiable goal for well-being. To address rising healthcare costs, innovative employers are investing in their workforce through Worksite Health Promotion programs [60]. Model 2 found that behaviors related to "health responsibility," "appreciation of life," and "healthy eating" were linked to sleep disturbances. People who prioritize health and nutrition tend to sleep better than those who completely neglect it, consistent with the findings of this study [61] According to Merrill et al. (2011) [62], participation in health pro- grams leads to improvements in exercise, diet, and stable sleep patterns, enhancing health and life satisfaction. Such programs help employees develop healthy behaviors, emphasizing the role of workplace nurses in promoting well-being. Weitzer et al. (2021) [63] observed that practicing "appreciation of life" behaviors such as maintaining an optimistic outlook, can improve sleep quality and reduce disturbances, consistent with this study's findings. The questionnaire on "Life appreciation behaviors" included items like "maintaining a smile or laughter every day" and "appreciating people, events, and nature." Thus, fostering a positive mindset, frequent smiles, self-appreciation, and spending time outdoors are recommended to lower sleep disturbance risks. This study supports findings by Cao et al. (2020) [64] and Castro-Diehl et al. (2018) [65] that "healthy eating behavior" is linked to sleep disturbances. Castro-Diehl et al. (2018) [65] also found that a healthy diet can promote better sleep duration and reduce insomnia 399
risk. It is recommended to consult nutritionists to adjust employee cafeteria meals toward a Mediterranean diet (featuring healthy fats, moderate fish and poultry, limited red meat) and whole grains [64,66], promoting healthy eating habits among employees. "Health responsibility" involves a proactive attitude towards one's well-being, including seeking health information and monitoring physical and mental states. Higher "health responsibility" scores were associated with fewer sleep disturbances, likely due to better self-management. Future studies could explore this relationship further.
Although previous research Xie et al. (2021) [67] and Bjornsdottir et al. (2024) [68] indicates that regular exercise improves sleep quality and reduces insomnia, this study did not find a clear link, possibly because most participants did not engage in regular exercise. Encouraging regular physical activity, especially for shift workers, could improve sleep quality [69,70]. Future initiatives might include smart bands, stair-climbing campaigns, and accessible workplace exercise options, with follow-up studies to confirm exercise's benefits for sleep disturbances. In Table 3, both groups with and without sleep disturbances showed low exercise scores. Future studies could explore the potential of encouraging regular physical activity to examine its relationship with sleep disturbances [61]

Response 17: Thank you for pointing this out. We agree with this comment. We have revised text in the manuscript, please see page number 11, lines 385~407.

Comments 18: Don’t use numbers in conclusion

Response 18: Thank you for pointing this out. We agree with this comment. We have revised text in the manuscript, please see page number 12, lines 455~ 464.

Reviewer 4 Report

Comments and Suggestions for Authors

The section Study Setting should explain how the sample was selected ( Purposive sampling....Convenience sampling. ..)

Author Response

Dear reviewer,

Thank you so much for the suggestion and we followed your comment and revised the paper again. Please see the responses as below.

Comment 1: The section Study Setting should explain how the sample was selected ( Purposive sampling....Convenience sampling. ..)

Response 1: Thank you for pointing this out. We used purposive sampling. We have revised text in the manuscript, please see page 3, line 136.

[The study participants were selected using purposive sampling and conducted in the traditional industrial factory. Following discussions with department management regarding safety considerations, the research was confined to plant 1.]